

# Influence of atmospheric deposition on biogeochemical cycles in an oligotrophic ocean system

France Van Wambeke[1], Vincent Taillandier[2], Karine Desboeufs[3], Elvira Pulido-Villena[1], Julie Dinasquet[4,5], Anja Engel[6], Emilio Marañón[7], Céline Ridame[8], Cécile Guieu[2]

[1] Aix-Marseille Université, CNRS/INSU, Université de Toulon, IRD, Mediterranean Institute of Oceanography (MIO) UM 110, 13288, Marseille, France

[2] CNRS, Sorbonne Université, Laboratoire d'Océanographie de Villefranche (LOV), UMR7093, 06230 Villefranche-sur-Mer, France

[3] LISA, UMR CNRS 7583, Université Paris-Est-Créteil, Université de Paris, Institut Pierre Simon Laplace (IPSL), Créteil, France

[4] Sorbonne Universités, Laboratoire d'Océanographie Microbienne (LOMIC), Observatoire Océanologique, 66650, Banyuls/mer

[5] Marine Biology Research Division, Scripps Institution of Oceanography, UCSD, La Jolla, USA

[6] GEOMAR – Helmholtz-Centre for Ocean Research, Kiel, Germany

[7] Department of Ecology and Animal Biology, Universidade de Vigo, 36310 Vigo, Spain

[8] Sorbonne University/CNRS/IRD/MNHN, LOCEAN: Laboratoire d'Océanographie et du Climat: Expérimentation et Approches Numériques, UMR 7159, 4 Place Jussieu – 75252 Paris Cedex 05, France

*Correspondence to*: France Van Wambeke (france.van-wambeke@mio.osupytheas.fr) and

Cécile Guieu (guieu@obs-vlfr.fr)

**Abstract.**

The surface mixed layer (ML) in the Mediterranean Sea is a well stratified domain characterized by low macro-nutrient and low chlorophyll content, during almost 6 months of the year. Nutrient dynamics in the ML depend on allochthonous inputs, through atmospheric

deposition and on biological recycling. Here we characterize the biogeochemical cycling of N in the ML by combining simultaneous *in situ* measurements of atmospheric deposition, nutrients, hydrological conditions, primary production, heterotrophic prokaryotic production, $N_2$ fixation and leucine aminopeptidase activity. The measurements were conducted along a 4300 km transect across the central and western open Mediterranean Sea in spring 2017. Dry

deposition was measured on a continuous basis while two wet deposition events were sampled, one in the Ionian Sea and one in the Algerian basin. Along the transect, N budgets were computed to compare sources and sinks of N in the mixed layer. On average,



phytoplankton N demand was 2.9 fold higher (range 1.5-8.1) than heterotrophic prokaryotic N demand. *In situ* leucine aminopeptidase activity contributed from 14 to 66 % of heterotrophic

prokaryotic N demand, and $N_2$ fixation rate represented 1 to 4.5 % of the phytoplankton N demand. Dry atmospheric deposition of inorganic nitrogen, estimated from dry deposition of (nitrate+ammonium) in aerosols, was higher than $N_2$ fixation rates in the ML (on average 4.8 fold). The dry atmospheric input of inorganic N represented a highly variable proportion of biological N demand in the ML, 10-82% for heterotrophic prokaryotes and 1-30% for

phytoplankton. Stations visited for several days allowed following the evolution of biogeochemical properties in the ML and within the nutrient depleted layers. At the site in the Algerian Basin and on a basis of high frequency sampling of CTD casts before and after a wet dust deposition event, different scenarios were considered to explain a delayed appearance of peaks in dissolved inorganic phosphate in comparison to nitrate within the ML. After the rain,

nitrate was higher in the ML than in the nutrient depleted layer below. Estimates of nutrient transfer from the ML to the nutrient depleted layer could explain ⅓ of the nitrate fate out of the ML. Luxury consumption of P by heterotrophic prokaryotes, further transferred in the microbial food web, and remineralized by grazers, is one explanation for the delayed phosphate peak of DIP. The second explanation is a transfer from ML to the nutrient depleted

layer below through adsorption/desorption processes on particles. Phytoplankton did not benefit directly from atmospheric inputs in the ML, probably due to a high competition with heterotrophic prokaryotes, also limited by N and P availability at the time of this study. Primary producers, in competition for nutrients with heterotrophic prokaryotes, decreased their production after the rain, recovering their initial state of activity after 2 days lag in the

vicinity of the deep chlorophyll maximum layer.

## 1. Introduction

The Mediterranean Sea (MS) is a semi-enclosed basin characterized by very short ventilation and residence times with respect to its own thermohaline circulation (Mermex Group, 2011).

In terms of biogeochemistry, MS is characterized by a long summer stratification period. It is a low nutrient, low chlorophyll (LNLC) system with a general west-to-east gradient of increasing oligotrophy, and an overall phosphorus (P) deficit compared to nitrogen (N) (Mermex Group, 2011). This is evidenced by higher N/P than Redfield ratios of inorganic nutrients in deep layers, increasing to the east (Krom et al., 2010).



The nature of the relationship between photoautotrophic unicellular organisms and heterotrophic prokaryotes (competition or commensalism) is affected by the balance of light and nutrients experienced by phytoplankton as well as possible inputs of organic matter from river runoff or atmospheric deposition. Phytoplankton generally experience P, N, or NP limitations (Thingstad et al., 2005 ; Tanaka et al., 2011, Richon et al., 2018), whereas

heterotrophic prokaryotes are usually P or P+labile carbon co-limited (Sala et al., 2002, Van Wambeke et al., 2002, Céa et al., 2014).

The MS continuously receives anthropogenic aerosols originating from industrial and domestic activities from all around the basin and other parts of Europe and pulsed natural inputs from the Sahara. It is thus, a natural LNLC laboratory well adapted to explore the role

of ocean–atmosphere exchanges of particles and gases on marine biogeochemical cycles. Parameterization and representation of the key processes brought into play by atmospheric deposition in MS must take into account the role of pulsed atmospheric inputs (Guieu et al., 2014a). Recent studies describe annual records of atmospheric deposition of trace metals and inorganic macronutrients (N, P) obtained in several atmospheric stations located around the

MS (Markaki et al., 2010; Guieu and Ridame, in press; Desboeufs, in press). All records denote the pulsed and highly variable inputs. Recent models and observations show that atmospheric deposition of organic matter (OM) is also highly variable and that annual atmospheric inputs of OM are of the same order of magnitude as river inputs (Djaoudi et al., 2017, Kanakidou et al., 2018; Kanakidou et al., 2020). Moreover, the sum of atmospheric

inputs of nitrate, ammonium and soluble organic nitrogen has been shown to be equivalent or higher than those of $N_2$ fixation rates (Sandroni et al., 2007), although inorganic atmospheric N inputs alone might also be higher than $N_2$ fixation rates (Bonnet et al., 2011).

Aerosol amendments in bottles, minicosms, or mesocosms have been widely used to study trophic transfer and potential export, as they represent to some extent a 'simplified' plankton

trophic web studied in the vertical dimension (i.e. Guieu et al. 2010; Herut et al., 2016). Both diversity and functioning of various biological compartments have been shown to be impacted by aerosol additions in different tested waters of the Mediterranean Sea (Guieu and Ridame, in press, and synthesis Figure 3 therein). Differences in the biological responses have been observed, depending on the mode of deposition (wet or dry) mimicked, the type of aerosols

(natural or anthropogenic) used and the *in situ* biogeochemical conditions at the time of the experiment (Guieu and Ridame, in press).



Organic carbon from aerosols is partly soluble, and this soluble fraction is partly available to marine heterotrophic prokaryotes (Djaoudi et al., 2020). Heterotrophic prokaryotes have the metabolic capacities to respond quickly to aerosol deposition through growth and changes in

community composition (Rahav et al., 2016; Pulido-Villena et al., 2008; 2014), while the phytoplankton community respond slower or not at all (Guieu and Ridame, in press, and reference therein).

Owing to the intrinsic limitations, which vary depending on the size and design of enclosures, i.e. without the representation of higher trophic levels, in absence of turbulent mixing that

limited exchanges to diffusion, wall effects, such experiments cannot fully simulate *in situ* conditions (Guieu and Ridame, in press). Thus, it is still necessary to acquire *in situ* observations of the the consequences of aerosol deposition on biogeochemical cycles. However, *in situ* studies of aerosol deposition are scarce and require high frequency sampling to follow the effects of deposition on biogeochemical processes. For instance, the biological

response to a Saharan dust event of 2.6 g m$^{-2}$ recorded at a land station (Cap Ferrat) was detected 4 days after by only one water column sampling offshore (at DYFAMED observation site), and analysis showed increases in heterotrophic prokaryotic respiration and abundances by comparison to a sampling date done 16 days before this dust event (Pulido-Villena et al., 2008). In the Israelian coast, a moderate natural dust storm (1.05 mg L$^{-1}$ over 5

m depth) was followed every 12 h during 5 days (Rahav et al., 2016). These authors observed rapid changes which may have been missed without high frequency sampling. They reported a decrease in picophytoplankton abundance, an increase in heterotrophic prokaryotes abundance, as well as slight increase in primary production (25%) and heterotrophic prokaryotic production (15%).

Hence, it is necessary to organize sampling surveys with adaptative strategies to follow aerosol deposition events *in situ* and their impacts on biogeochemical processes, especially in the open waters of the summer stratified and nutrient limited MS. The objectives of the PEACETIME project were to study fundamental processes and their interactions at the ocean–atmosphere interface after atmospheric deposition (especially of Saharan dust) in the

Mediterranean Sea, and how these processes impact the functioning of the pelagic ecosystem (Guieu et al., 2020).

As atmospheric deposition affects primarily the surface mixed layer (ML), the present study focuses on the upper part of the nutrient depleted layer that extends down to the nutriclines (as defined by Du et al., 2017). During the stratification period, the concentrations of nitrate and





phosphate within the ML are often below quantification limits of standard methods. However, nanomolar concentrations of nitrate (and phosphate) can now be assessed accurately through Long Waveguide Capillary Cell (LWCC) technique, which allows to measure fine gradients inside nutrient depleted layers in the MS (Djaoudi et al., 2018).

The aims of the present study were to assess the impact of atmospheric nutrient deposition on
biogeochemical processes and fluxes in the open sea during a cruise in the MS. For this i) we estimated nanomolar variations of nitrate concentration in the surface mixed layer (ML) facing variable inputs of dry and wet aerosols deposition ii) we compared the aerosols-derived N inputs to the ML with biological activities: primary production, heterotrophic prokarytic production, $N_2$ fixation and ectoenzymatic activity (leucine aminopeptidase). We studied
these N budgets along a 13 stations transect crossing the Algerian Basin, Tyrrhenian Sea and the Ionian Sea where dry atmospheric deposition was continuously measured on board together with seawater biogeochemical and biological characteristics. We finally focused on a high frequency sampling of a wet deposition event that occurred in the western Algerian Basin, during which we investigated the evolution of biogeochemical fluxes of both N and P
and microbial activities.

## 2. Materials and Methods

### 2.1 Sampling strategy and measured parameters

The PEACETIME cruise (doi.org/10.17600/15000900) was conducted in the Mediterranean
Sea, from May to June 2017, along a transect extending from the Western Basin to the center of the Ionian Sea (Fig. 1). For details on the cruise strategy, see Guieu et al. (2020). Short duration stations (< 8 h, 15 stations named ST1 to ST10, Fig. 1) and long duration sites (5 days, 3 sites named TYR, ION and FAST) were occupied. Chemical composition of aerosols was quantified by continuous sampling along the whole transect. In addition, two rain events
were sampled (Fu et al., this issue, in prep. b), one on the 29[th] of May at ION station, and a second one, a dust wet deposition event, at FAST site on the 5[th] of June.

At least 3 CTD casts were conducted at each short station. One focused on the epipelagic layer (0-250 m), and the second one on the whole water column. Both were sampled with a standard, classical-CTD rosette equipped with a sampling system of 24 Niskin bottles (12 L),
and a Sea-Bird SBE9 underwater unit equipped with pressure, temperature (SBE3), conductivity (SBE4), chlorophyll fluorescence (Chelsea Acquatracka) and oxygen (SBE43) sensors. The third cast, from surface to bottom was performed under 'trace metal clean



conditions' using a second instrumental package including a titanium rosette (called TMC-rosette) mounted on a Kevlar cable and equipped with Go-Flo bottles that were sampled in a
dedicated clean lab container. The long sites were abbreviated as TYR (situated in the center of the Tyrrhenian Basin), ION (in the center of the Ionian Basin), FAST (in the western Algerian Basin). These 3 sites were selected using satellite imagery, altimetry and Lagrangian diagnostics and forecasted events of rain (Guieu et al., 2020). At these sites, repeated casts were performed during at least 5 days, alternating CTD- and TMC- rosettes. TYR site was
occupied from May 17, 05:08 to May 21, 15:59 and ION site from May 24, 18:02 to May 29, 8:25. FAST site was occupied from June 2, 20:24 to June 7, 23:25 and then from June 8, 21:06 to June 9, 00:16 (all times in local time). The succession of CTD casts at FAST site are numbered in days relative to a rain event collected on board the ship that ended on June 5, 3:04. The first cast of the series was sampled 2.3 days before that rain, and the last one 2 days
after. The FAST site was occupied a second time after the study of ST10 (3.8 days after the rain event).

Primary production (PP), prokaryotic heterotrophic production (BP), heterotrophic prokaryotic abundances (hprok), ectoenzymatic activities (leucine aminopeptidase (LAP) and alkaline phosphatase (AP)), were determined on water samples collected with the classical
CTD-rosette. Inorganic nutrients, dissolved organic nitrogen (DON) and phosphorus (DOP) were measured on water samples collected by the TMC-rosette. LAP and AP were determined on two layers in the epipelagic waters (5 m depth and deep chlorophyll maximum (DCM)) at the short stations as well as at ION and TYR stations. In addition, for 4 profiles at FAST, LAP and AP were determined at 4 depths between 0 and 20 m to get insight on the variability
within the ML.

## 2.2 Analytical methods and fluxes calculations
### 2.2.1 Nutrients in the atmosphere

Description of the atmospheric deposition strategy is detailed in Desboeufs et al. (this issue, in
prep). In summary, the total suspended aerosol particles (TSP inlet) were continuously collected during the campaign, either by filtration units on adapted membranes for off-line chemical analysis or by aerosol water extraction unit, a Particle-into-Liquid-Sampler (PILS, Orsini et al., 2003), for on-line analyses of major water soluble ions. Moreover, two wet deposition events were sampled on board the ship during the cruise (details in Fu et al., this
issue, in prep. b), one at ION site, one at FAST site. At FAST, casts were performed before,





and after the rain event at a high frequency with a fine definition of the sampling within the mixed layer, between surface and 20 m depth. This was done also at ION but with a lower frequency. Also, a strong dust deposition event, with intense wet deposition, likely occurred over the Tyrrhenian Sea around May 11-12th, a few days before our arrival on site

(Desboeufs et al., this issue, in prep.). Using particulate aluminum inventory in the water column, the dust deposition was estimated to be 5.5-5.9 g m$^{-2}$ at ST5, 0.8-1.6 at TYR and 1.6-1.9 at ST6 (Bressac et al., this issue, in prep.).

Nitrate and ammonium concentrations in aerosols, abbreviated in the text as NO3 and NH4, respectively, (nitrite concentrations were under analytical detection limits) were continuously

analyzed on board from May 13 by PILS sampling coupled on-line with a double way ion chromatography (PILS-IC, Metrohm, model 850 Professional IC with Metrosep A Supp 7 column for anions measurements and Metrosep C4 column for cation measurements; Fu et al., this issue, in prep a). Inorganic phosphate (DIP) concentrations were estimated from total phosphorus concentrations in particulate aerosols on filters since it was generally under

detection limits using the PILS-IC technique. NO3, NH4 and DIP were also determined by ion chromatography in the dissolved fraction of the 2 rain samples.

Atmospheric dry deposition fluxes of inorganic nitrogen were estimated by multiplying the aerosol concentrations of NO3 and NH4 by the dry settling velocities of N-bearing aerosols, i.e. 0.21 and 1 cm s$^{-1}$ for NH4 and NO3, respectively (Kouvarakis et al., 2001). The

concentrations used were averages from PILS-IC analysis obtained during the occupation of each short station, and averages between two successive casts during site occupations, except for ST1 where concentrations were issued from filter by IC analyses after water extraction. This period lasted from 0.13 to 0.66 days for short stations and from 0.41 to 1.21 days for sites. Atmospheric dry deposition of inorganic P was estimated using the total P particulate

aerosol concentration multiplied by a dry settling velocity of 1 cm s$^{-1}$ except 3 cm s$^{-1}$ at FAST site, as this value is more adapted for Saharan events (Izquierdo et al., 2012), hence this value is likely an upper range for DIP. At each short station, total P was determined from the filter sampled during the period of the short station and this period of filter collection ranged 0.28 - 1.15 days according to the stations. At the 3 sites, we used filters collecting aerosols during

periods including the CTD casts sampling date, when possible. The wet deposition fluxes were estimated from the measured concentrations in the filtrate phase of rain (0.2 µm), the rain volume and the surface of the rain funnel (Fu et al., this issue, in prep. b).



### 2.2.2 Nutrients in the water column

Seawater samples for standard dissolved nutrient analysis were filtered online (< 0.2 μm, Sartorius Sartrobran-P-capsule with a 0.45 μm prefilter and a 0.2 μm final filter) directly from the GoFLo bottles (TMC-rosette). Samples collected in acid-washed polyethylene bottles were immediately analyzed on board. Micromolar nitrate + nitrite (NOx) and DIP were determined using a segmented flow analyzer (AAIII HR SealAnalytical©) following Aminot

and Kérouel (2007) with a limit of quantification of 0.050 μM for NOx and 0.020 μM for DIP. Samples for the determination of nanomolar concentrations of dissolved nutrients were collected in HDPE bottles previously cleaned with supra-pure HCl. For NOx (considered as NO3 as nitrite fraction was mostly negligible), samples were acidified to pH 1 inside the clean van and analyzed back to the laboratory using a spectrometric method in the visible at 540

nm, with a 1 m LWCC (Louis et al., 2015). The limit of detection was 6 nM, the limit of quantification was 9 nM and the reproducibility was 8.5%. DIP was analyzed immediately after sampling using the LWCC method after Pulido-Villena et al. (2010), with a limit of detection of 1 nM. Total dissolved phosphorus (TDP) and nitrogen (TDN) were measured using the segmented flow analyzer technique after high-temperature (120 °C) persulfate wet

oxidation mineralization (Pujo-Pay and Raimbault, 1994). DOP (DON) was obtained as the difference between TDP (TDN) and DIP (NOx). Labile DOP (L-DOP) was determined as 31% DOP (Pulido-Villena et al., this issue, in prep.).

Total hydrolysable amino acids (TAA) were determined as described in detail in Van Wambeke et al. (2020). Briefly 1 ml of sample were hydrolyzed at 100°C for 20 h with 1 ml

of 30% HCl and then neutralized by acid evaporation. Samples were analyzed by high performance liquid chromatography according to Dittmar et al. (2009) protocols.

### 2.2.3 Biological stocks and fluxes in the epipelagic waters

For the enumeration of autotrophic prokaryotic and eukaryotic cells, heterotrophic

prokaryotes (hprok) and heterotrophic nanoflagellates (HNF) by flow cytometry, subsamples (4.5 mL) were fixed with glutaraldehyde grade I 25% (1% final concentration), flash frozen and stored at -80 °C until analysis. Counts were performed on a FACSCanto II flow cytometer (Becton Dickinson). The separation of different autotrophic populations was based on their scattering and fluorescence signals according to Marie et al. (2000). For the

enumeration of hprok (Gasol and Del Giorgio, 2000), cells were stained with SYBR Green I



(Invitrogen – Molecular Probes). HNF staining was performed with SYBR Green I as described in Christaki et al. (2011). All cell abundances were determined from the flow rate, which was calculated with TruCount beads (BD biosciences).

Particulate primary production (PP) was determined on 6 layers from the shallow (0-250 m) casts sampled before sun rise. Samples were inoculated with $^{14}$C-bicarbonate and incubated in on-deck incubators refrigerated with running surface seawater and equipped with various blue screens to simulate different irradiance levels. After incubations during 24 h, samples were filtered through 0.2 polycarbonate filters and treated for liquid scintillation measurement as described in detail in Marañón et al. (2020). A temperature correction was applied as

explained in Marañón et al. (2020). $N_2$ fixation rates (N2fix) were determined as described in Ridame et al. (this issue, in prep) using 2.3 L of unfiltered seawater (acid-washed polycarbonate bottles) enriched with $^{15}$N2 gas (99 atom% 15N) to obtain a final enrichment of about 10 atom% excess. Incubations for N2fix were conducted under the same temperature and irradiance as the corresponding PP incubations.

To calculate prokaryotic heterotrophic prokaryotic production (BP) epipelagic layers (0-250 m) were incubated with tritiated leucine using the microcentrifuge technique as detailed in Van Wambeke et al. (2020). We used the empirical conversion factor of 1.5 ng C per pmol of incorporated leucine according to Kirchman (1993). Indeed, isotope dilution was negligible under these saturating concentrations as checked occasionally with concentration kinetics. As

we used only 2 temperature controlled dark-incubators on board, a temperature correction was applied as explained in Van Wambeke et al. (2020). Ectoenzymatic activities were measured fluorometrically, using fluorogenic model substrates that were L-leucine-7-amido-4-methyl-coumarin (Leu-MCA) and 4 methylumbelliferyl – phosphate (MUF-P) to track aminopeptidase activity (LAP), and alkaline phosphatase activity (AP), respectively, are

described in details in Van Wambeke et al. (2020). Briefly the release of MCA from Leu-MCA and MUF from MUF-P were followed by measuring increase of fluorescence in the dark (exc/em 380/440 nm for MCA and 365/450 nm for MUF, wavelength width 5 nm) in a VARIOSCAN LUX microplate reader. Fluorogenic substrates were added at varying concentrations in 2 ml wells (0.025, 0.05, 0.1, 0.25, 0.5 and 1 µM) in duplicate. From varying

velocities obtained, we determined the parameters $V_{max}$ (maximum hydrolysis velocity) and Km (Michaelis-Menten constant that reflects enzyme affinity for the substrate) by fitting the data using a non-linear regression on the following equation:





V = Vmax × S/(Km + S)

where V is the hydrolysis rate and S the fluorogenic substrate concentration added. TAA and

L-DOP were used as substrates to determine LAP and AP *in situ* activities from Michaelis-

Menten equations (Van Wambeke et al., 2020, Pulido-Villena et al, this issue, in prep.).

### 2.3 Enrichment experiments

At the three long duration sites, enrichment experiments were performed using seawater from

5 m depth to assess factors limiting BP in the surface mixed layer. The date of sampling for

these experiments [FAST (June 2, 22:00), TYR (May 16, 20:00) and ION (May 25, 20:00)]

was before the rain event occurring at FAST and at ION. Eight series of triplicate 60 mL

polycarbonate bottles were filled with unfiltered seawater and amended as follows: C : no

enrichment, N: +1 µM $NO_3$ + 1 µM $NH_4$; P: + 0.2 µM DIP; G: + 10 µM C-glucose; NP: N +

P; NG : N + G; PG : P + G; NPG: N + P + G. After 48 h incubation in the dark at *in situ*

temperature, BP was determined in the 24 bottles (as described in previous section).

### 2.4. Vertical partition of the epipelagic layer

The ML can be in a state of stirring and the density and seawater properties are homogeneous,

or in a state of rest with homogeneous density but vertical variations of the non-conservative

components (such as nutrients); depending on momentum and heat fluxes at the interface with

the atmosphere that determine the strength and vertical extent of mixing (Brainerd and Gregg,

1995). In the present study, the mixed layer depth (MLD) was estimated indirectly using the

shape of CTD profiles, in absence of concomitant turbulence measurements. Classical

methods to evaluate MLD from CTD profiles are based on thresholds of vertical density gap

(de Boyer Montegut et al., 2004; D'Ortenzio et al., 2005) or on the vertical extension of a

fixed buoyancy content (Moutin and Prieur, 2012). As discussed in Gardner et al. (1995), the

choice of criterion is sensitive to discriminate the subtle changes from active mixing to rest,

with consequences in the representative time scales of MLD estimates. In the present study,

the ML were shallow (10 - 20 m), rapidly activated by mechanical effects of wind, and

sampled at high frequency (some hours at long duration stations). An approach based on

buoyancy criterion has been preferred to better resolve short term fluctuations in the mixing

state, and MLD was determined as the depth where the residual mass content is equal to 1 kg

$m^{-2}$, with an error of estimation of 0.5 m relative to the vertical resolution of the profile (1 m).





In the present case of shallow ML in LNLC systems, the nutrient depleted layer comprises the
ML and a layer below, referred hereafter as 'NDLb' (b for bottom or base) when examining
NO3 distribution and PDLb when considering DIP distributions. This layer vertically extends
between the MLD and the nitracline (phosphacline) depth (Fig. 2). The latter interface is
estimated by the depth of the NO3 (DIP) depletion density, which is the deepest isopycnal at

which micromolar NO3 (DIP) concentration is zero (Kamykowski and Zentara, 1986; Omand
and Mahadevan, 2015). The NO3 (DIP) depletion density is estimated at every discrete
profile of micromolar NO3 (DIP) concentrations by the intercept of the regression line
reported in a nutrient-density diagram.

There are various mechanisms, dynamical or biological, that can trigger exchanges of

nutrients between the ML and NDLb (PDLb). Under the hypothesis of vertical (one-
dimensional) regimes, the processes of exchange are twice, by diffusion or advection (e.g. Du
et al., 2017). The flux of nutrient can be expressed as:

$$F_{NO3} = F_{DIF} + F_{ADV}$$

The diffusive flux $F_{DIF}$ is expressed by the gradient of nutrient concentration times a vertical

diffusivity coefficient $K_z$ as:

$$F_{DIF} = K_z . (NO3_{ML} - NO3_{NDLb}) / MLD$$

The typical magnitude of $K_z$ in the surface layers of the PEACETIME stations is assessed to
$10^{-5} \, m^2 \, s^{-1}$, as discussed in Taillandier et al. (2020).

The advective flux $F_{ADV}$ is expressed by the variation of nutrient concentration across the ML

times a temporal variation of MLD, as:

$$F_{ADV} = (NO3_{ML} - NO3_{NDLb}) .dMLD / dt$$

With shallow ML (10 - 20 m) primarily influenced by wind bursts that lead to intermittent
MLD variations of some meters per day ($10^{-5} \, m \, s^{-1}$), the advective fluxes provide transient
exchanges of one order of magnitude larger than well-established diffusive fluxes. In

consequence, at the time scale of a rain event and the associated rapid variations of MLD, the
atmospheric input of nutrients is preferentially exported below the ML by advection rather
than by diffusion. In other terms, under the hypothesis of non-stationary regimes due to rapid
changes in atmospheric conditions (that control both mixing state of the interface and aerosol
nutrient inputs) we assume that vertical advection is the main process of exchange.

At short stations and long duration sites, the term $NO3_{ML} - NO3_{NDLb}$ can be inferred by
difference of nanomolar (LWCC) concentrations inside NDLb relative to the ones inside ML.
Since advective flux estimates require the evolution of MLD, they are only accessible at the





long duration sites. At short stations, and in absence of characterization of advective fluxes, only a qualitative assessment of nutrient fluxes across ML can be drawn. When $NO3_{ML} -$

$NO3_{NDLb} < 0$, the NDLb is supplied in nutrients across the nutricline, and then possibly transferred inside the ML, meaning that ML nutrients are impacted by inputs from below. When $NO3_{ML} - NO3_{NDLb} > 0$, the ML is supplied in nutrients from above and exported into the NDLb. Vertical distributions of DIP, along the longitudinal transect, are described in detail in a companion paper (Pulido-Villena et al., this issue, in prep.). In this study, we will

focus on phosphate exchanges between ML and PDLb layers at the high-frequency sampled FAST site and relationships with NO3 distribution.

### 2.5 Budget from the metabolic fluxes

Trapezoidal integration was used to integrate BP, PP and N2fix within the ML. The biological

activity at surface was considered equal to that of the first layer sampled (around 5 m depth at the short stations, 1 m depth at the FAST site). When the MLD was not sampled, the volumetric activity at this depth was linearly interpolated between 2 closest data above and below the MLD.

We used an approach similar to Hoppe et al. (1993) to compute *in situ* hydrolysis rates for

LAP and AP. We assumed that total amino acids (TAA) could be representative of dissolved proteins. *In situ* hydrolysis rates of LAP and AP were determined using molar concentrations of TAA and L-DOP, respectively used as substrate concentration in the Michaelis-Menten kinetics. For LAP, the transformation of *in situ* rates expressed in nmol TAA hydrolysed $L^{-1}$ $h^{-1}$ were then transformed in nitrogen units using N per mole TAA, as the molar distributions

of TAA were available. Integrated *in situ* LAP hydrolysis rates were calculated assuming the Michaelis-Menten parameters Vm and Km obtained at 5 m depth to be representative of the whole ML, and thus an average *in situ* volumetric LAP flux in ML was obtained by combining average TAA concentrations in the ML with these kinetic parameters, and then this volumetric rate was simply multiplied by the MLD. Daily BP, AP and LAP integrated

activities were calculated from hourly rates x 24. Assuming no direct excretion of either nitrogen or phosphorus, the quota C/N and C/P of cell demand is equivalent to cell biomass quotas. We used C/N ratios derived from Moreno and Martiny, 2018 (range 6-8, mean 7) for phytoplankton and from Nagata et al. (1986) for heterotrophic prokaryotes (range 6.2-8.4, mean 7.1). C/P of sorted cells (cyanobacteria, picophytoeukaryotes) in P depleted conditions

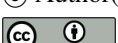



ranged 107 to 161 (Martiny et al., 2013), and we considered a mean of 130 for phytoplankton. A value of 100 was used for heterotrophic prokaryotes (Godwin and Cotner, 2015).

## 3. Results

**3.1 Nutrients pattern and biological fluxes along the PEACETIME transect**

The ML occupied the first 20 m with an excursion between 9 m in the Ionian Sea (ION site) and 21 m in the Gulf of Lions (ST1, Table S1, Fig. 3). The nitracline was shallow in the Provençal Basin (50 – 60 m), dropping down in the Eastern Algerian and Tyrrhenian at 70 m; it was the even deeper in the Western Algerian and Ionian (80 – 90 m). Some anomalies

appeared on this progressive deepening of the nitracline from northwest to east and southwest, likely due to mesoscale activity (end of FAST and ST10), or a situation close to the continental shelf (ST3). Mean NDLb $NO_3$ concentrations ranged from detection limits using LWCC technique (9 nM) to 116 nM (Table S1, Fig. 4). In the ML, mean $NO_3$ concentration ranged from 9 to 135 nM. Note that mean $NO_3$ concentration was not systematically low

within the ML, particularly at FAST, ION and ST10. For the stations 5, TYR, 6, 7, ION25 May, FAST+1.05, FAST+2.1 (hereafter 'group 1'), the nitrate concentrations were low (below 50 nM) in both ML and NDLb layers, with weak differences between the two layers. For the ST8 and the casts done before the rain event at FAST (hereafter 'group 2'), the nutrient concentrations were moderate (50 - 80 nM) both in ML and NDLb with weak

difference between the two layers, indicating small gradients between ML and NDLb and thus, likely little instantaneous exchanges (at the time of the cast). For ST9, higher nitrate concentrations were measured in both ML and NDLb (> 80 nM) but the weak positive differences (< 20 nM) between the two layers also likely indicate weak or null exchanges between the two layers at the time of the cast (hereafter 'group 3'). For ST10, ION27 and

ION29 May, and FAST+0.25, +0.5 and +3.8 (hereafter 'group 4'), high and moderate nitrate concentrations were measured in the ML and NDLb, respectively, with a large positive difference (> 20 nM) between both layers.

Concerning DIP, nanomolar data indicated the presence of positive vertical gradients above the phosphacline, in all stations of the transect, so that DIP in the ML was always lower than

in the PDLb above except at ION and ST6 (where no gradient was detected) (Pulido-Villena et al. this issue, in prep.).



The vertical distribution of PP and BP are described in Marañon et al. (2020) for the short
stations. Briefly, PP exhibited a deep maximum close to the DCM depth or slightly above
whereas vertical distribution of BP generally showed 2 maxima, one within the mixed layer,
and a second one close to the DCM. Integrated PP (Table 1) ranged from 138 (TYR17 May)
to 284 (SD1) mg C m$^{-2}$ d$^{-1}$ (mean ± sd: 194 ± 48 mg C m$^{-2}$ d$^{-1}$). Integrated BP (0-200 m)
ranged from 44 (ION27may) to 113 (FAST+0.53) mg C m$^{-2}$ d$^{-1}$ (mean 70 ± 20 mg C m$^{-2}$ d$^{-1}$).
The typical states in absence of rain event follow the west-east gradient of MS oligotrophy
detected by ocean color (see Fig. 8 in Guieu et al., 2020), from 'chlorophyll-rich' waters
encountered along the Algerian basin until 'chlorophyll-poor' waters in the Tyrrhenian Sea
and 'ultra-poor' surface water in the Ionian Sea.
VmLAP at 5 m depth ranged from 0.24 to 0.56 nmol MCA-leu hydrolyzed l$^{-1}$ h$^{-1}$, and Km
LAP ranged from 0.12 to 1.29 µM. The mean TAA within the ML ranged 0.17 to 0.28 µM.
From these 3 series of parameters, we derived mean *in situ* LAP hydrolysis rate within the
ML, which ranged from 0.07 to 0.29 nmol N l$^{-1}$ h$^{-1}$ (results not shown but detailed in Van
Wambeke et al., 2020).

**3.2 N budgets and fluxes at short stations**

Focusing on the surface mixed layer, biological rates, all expressed in N units, were
calculated as described in Material and Methods section and compared among the short
stations ST1 to ST10 (Table 2). Phytoplankton N demand (phytoN demand) was the largest
rate, followed by heterotrophic prokaryotic N demand (hprokN demand), *in situ* LAP
hydrolysis rates and N$_2$ fixation. On average, phytoN demand was 2.9 fold higher than
hprokN demand, however with a large variability (min x1.5, max x8.1). LAP hydrolysis rates
represented between 14 and 66 % of the hprokN demand, N$_2$ fixation rates represented from 1
to 4.5% of the phytoN demand.
Dry atmospheric deposition of inorganic N (DIN=NO3+NH4) was dominatedby NO3, which
represented on average 79% of the total inorganic N (Table 2). The estimated inputs of DIN
from dry deposition ranged from 17 to 40 µmol N m$^{-2}$ d$^{-1}$ with highest fluxes (> 35 µmol N m$^{-2}$ d$^{-1}$)
at ST3 and ST9. Dry deposition of DIN was similar or higher than N$_2$ fixation rates in
the ML (from 1.3 fold to 11 fold higher, mean 4.8 fold higher). On average, dry deposition of
inorganic N represented 27% of the hprokN demand within the ML (range 10-82%), and 11%
of the phytoN demand within the ML (range 1-30%).



### 3.3 Biogeochemical evolution at the long station ION

The ION site was occupied from the 25 to the 29 May, and the rain event occurred just 3 hours before the last CTD cast. However, rain events in the area of the site have been also observed on the day of May 26. CTD casts dedicated to biogeochemical studies were separated by 24 h (biological fluxes) or 48 h (DIP and NO3). Thus the time sequence of nutrients at ION is given only by three profiles. The first profile (25 May) is 'flat', corresponding to fair weather conditions and shallow ML with low and homogeneous concentrations of NO3 in the ML and the NDLb layers (Fig. 4). In the interval with the second profile, a depression arrived and the ML started to deepen 13 h before the second cast (27 May). This cast was marked by high nitrate in the ML, which is portioned at 10 m due to the history of the MLD (Fig. 3). The mixing should have set up homogeneous ML, but wind conditions rose at 20 kt just at the time of the cast (Fig. 3). The interval with the third cast (29 May, sampled 3 hours after the rain) is marked by slight relaxation of weather depression, and the deepening of ML down to 20 m. On 29 May, NO3 decreased in both ML and NDLb but still, values were higher in ML. Given the remaining signature at 15 m and deepening of the ML, the two layers might have stayed isolated. However, the calculation of vertical advective fluxes between the layers showed a downward flux in the first interval 25-27 May (Fig. 4, Table S1) and an upward flux in the second interval (27-29 May).

DIP dry atmospheric deposition decreased from 383 and 385 nmol P m$^{-2}$ d$^{-1}$ on the 26 and the 28 May down to 185 nmol P m$^{-2}$ d$^{-1}$ the 30 May. The percentage of DOP included in rain was also an important P source representing 60% of total dissolved P deposition. DIN (NO3+NH4) dry atmospheric fluxes could be sampled and daily flux calculated more regularly: it increased from 24 to 34 µmol N m$^{-2}$ d$^{-1}$ from the 25 to the 28 May and decreased to 29 µmol N m$^{-2}$ d$^{-1}$ on the 29 May after the rain. The rain sample was associated with a rain front covering more than 5 000 km$^2$ around the sampling zone, and passing during the night between 28 and 29 June. The rain event collected on board at ION lasted 52 min (Table 3). The DIN wet deposition flux was much higher than DIP flux, resulting in a molar N/P inorganic ratio of 208. As only one cast was sampled after the rain, we could not analyze the fate of this wet event in particular its possible effect on biological stocks and fluxes as a function of time.

Due to the lack of a high frequency sampling, it was also particularly difficult to assess direct time evolution effects of dry atmospheric deposition at ION site. Nevertheless, it was clear from the casts sampled on the 27 and 29 May that this site was characteristic of group 4 (i.e.





higher NO3 in the ML than in the NDLb), indicating recent inputs from the atmosphere. The integrated BP increase within the ML indicates that hprok benefited more from the

atmospheric inputs than PP which decreased at 5 m depth (Fig. S1).

At ION site, ectoenzymatic activities were only sampled on the 25 May. Vm of LAP at 5 m (0.22 nmol N l$^{-1}$ h$^{-1}$) was mostly one of the lowest values recorded during the cruise (min-max range of 0.21-0.56 nmol N l$^{-1}$ h$^{-1}$ for the whole cruise) and Vm of AP showed the highest value (5.6 nmol P l$^{-1}$ h$^{-1}$, for a min-max range of 0.3-5.6 nmol P l$^{-1}$ h$^{-1}$ for the whole cruise).

PP integrated over the euphotic zone increased slightly from 211 (25 May) to 282 mg C m$^{-2}$ d$^{-1}$ (28 May), but due to changes in the MLD during the occupation of the ION site (range 11-21 m) this trend was not visible when integrating over the ML. Integrated BP (0-200 m) ranged 43-63 mg C m$^{-2}$ d-1 with the highest value on the 28 May. Integrated over the ML, BP slightly increased, from 7.5 to 10.3 mg C m$^{-2}$ d$^{-1}$ between the 25 and the 29 May. The profiles

of hprok and *Synechococcus* abundances showed no particular trend with time, with higher variations within the DCM (Fig. S1).

**3.4 N budgets and fluxes at the long station FAST**

During the occupation of the FAST site NO3 concentration in the ML showed a progressive

enrichment with a maximum few hours after the rain event and a smooth decline until the initial conditions were restored 2.3 days after the rain (Fig. 3). Afterwards, the nutrient dynamics was not consistent with this evolution as shown by the high concentrations measured at FAST+3.8. The same analysis can be drawn with respect to exchanges across the ML (Fig. 4, Table S1): NO3 flux remained low until the rain event, and then a large advective

flux is estimated relative to the enrichment of ML and the quick deepening of the ML, this flux declined during the relaxation to initial conditions.

Concerning atmospheric wet deposition, the rain event collected on board at FAST lasted 28 min (Table 3). Like for the rain sampled at ION site, DIN flux was much higher than DIP flux, with molar N/P inorganic ratio of 1439, and the percentage of DOP represented 44% of

total dissolved P deposition. The chemical composition indicates that the rain sample collected at FAST site was characteristic of a wet dust deposition (Fu et al., this issue, in prep. b) corresponding to a dust deposition of 12 mg m$^{-2}$. This is among the least intense dust deposition fluxes recorded in this area from a long time-series of deposition (Vincent et al., 2016). During FAST site occupation, two periods of rains occurred related to rain fronts

(affecting a domain of ~5000 km$^{2}$) moving eastward from Spain and North Africa regions:



between the evening of June 3 until 5 in the early morning (Desboeufs et al., this issue, in prep.). These two episodes were concomitant with a dust plume transported in altitude (between 1and 4 km), covering the half of western Mediterranean basin during June 4, and allowed below-cloud deposition of dust (as confirmed by LIDAR records, Desboeufs et al.,

this issue, in prep.). The rain event collected at FAST site occurred on board after a first night of wet deposition over the whole area and toward the end of the transport of the dust plume. The particulate flux of this collected rain sample was thus likely in the lower range of the wet dust deposition fluxes that had affected the whole area between 3 and 5 June. Indeed, the *in situ* measurements of a dust tracer in the water column (particulate aluminum) show that the

total export of particulate matter was rather of the order of ~55 mg m$^{-2}$ during the period instead of 12 mg m$^{-2}$ as seen in a single collected rain event, and even this value is probably underestimated due to the lack of sampling big dust particles sinking (Bressac et al., this issue, in prep.). After the rain, average daily dry DIN deposition decreased from 45 μmol N m$^{-2}$ d$^{-1}$ on the June 4 (4:00-17:00), to 27 μmol N m$^{-2}$ d$^{-1}$ on the following night (June 4 21:00 -

June 5 4:40) and 9.8 μmol N m$^{-2}$ d$^{-1}$ on June 5 (9:20-15:00).

The beginning of the sequence at FAST site (-2.3; -1.5; -0.25) was marked by moderate decreases of NO3 concentration but with equal concentrations in the ML and NDLb. This stage reached a steady state after 2 days of evolution (means of 59 and 56 nM NO3 at FAST-1.5 and FAST-0.25). Integrated stocks of NO3 within the ML reflected slight changes in the

ML depth (from 14 to 10 m during that time).

The rainfall (rain 2, Table 3) was associated with a strong wind burst and an abrupt mixing. The two closest CTD casts framing the rain were sampled 6 h before and 6 h after (FAST-0.25 and FAST+0.24) and the comparison between both casts showed a clear enrichment of the ML: mean NO3 increased from 56 to 93 nM and integrated stocks increased by 888 μmol

N m$^{-2}$. In addition, there was a clear difference in the mean concentrations between ML and NDLb layers (93 ± 15 vs 51 ± 7 nM, respectively, which is the highest difference between these 2 layers observed during the cruise), confirming that this ML enrichment could not be attributed to inputs from below.

The relaxation of this wind burst was progressive, with a continuous deepening of the ML.

The export of the nitrate from atmospheric input into the NDLb was maximum after the rain event (FAST+0.24). A much smaller value was recorded at FAST 2.1 and fluctuated not significantly differently than 0 in between.



The end of the sequence (FAST+3.8) was marked by the return of higher NO3 (135 nM)
content inside the ML. However this event can be isolated from the previous sequence and

more difficult to interpret as persistence timescales (about 2 days in the hypotheses) are over.
If the NO3 stocks inside ML were already increasing 6 h after the rain event, however, it was
not the case for DIP which showed similar ML-integrated data 6 h before and 6 h after the
rain (136 µmol m$^{-2}$) and then progressively increased up to 281 µmol P m$^{-2}$ at FAST+1 (1 day
after the rain).

Immediately after the rain, integrated PP (euphotic zone) decreased from FAST-0.9: 274 mg
C m$^{-2}$ d$^{-1}$ to FAST+0.07: 164 mg C m$^{-2}$ d$^{-1}$ and continued to decrease the following day
(FAST+1.07: 140 mg C m$^{-2}$ d$^{-1}$). It is only after 3.8 days after the rain that integrated PP was
of the same order of magnitude as before the rain. Such variations were mostly due to changes
in volumetric rates within the DCM depth (Fig. S2), as the activity did not change

significantly within the ML (28-33 mg C m$^{-2}$ d$^{-1}$, Fig. S2, Fig. 5). Integrated BP 0-200 m
showed an opposite trend compared to PP, and tended to increase after the rain: $86 \pm 3$ mg C
m$^{-2}$ d$^{-1}$ (n = 4) before, and an increase from 80 (FAST+0.07) to 113 mg C m$^{-2}$ d$^{-1}$ (FAST+0.5)
after. The trend of increase was slight, but also visible when considering only ML (12-15
before; 15-19 m$^{-2}$ d$^{-1}$ after). The abundances of picophytoplankton groups were mostly

varying around the DCM depth with peaks occurring 1-2 days after the rain (grey profiles, Fig
S3), in particular for prokaryotes (*Prochlorococcus*, *Synechococcus*). Heterotrophic
prokaryotes and nanoflagellate abundances slightly also increased within the DCM depth after
the rain.

**3.5 Enrichment experiments at long stations**
At TYR site, BP was significantly stimulated only after addition of 2 major elements, with a
greater response with PN and NPG combinations (Fig. 6). At ION site, BP was primarily
limited by P availability, as only combinations with P, single or in combination with other
elements, stimulated BP whereas all other combinations of enrichment did not stimulate BP

significantly compared to the control. At FAST site, BP was primarily limited by N
availability, as all combinations with N, single or in combination with other elements,
stimulated BP. However at this site it is likely that BP was also co-limited by 2 elements, as G
addition alone also stimulated BP. Note that the PN combination has induced a higher
stimulation of BP than other double combinations (NG or PG).






## 4. Discussion

In this study, *in-situ* characterization of the biogeochemical response to atmospheric deposition was studied for the surface mixed layer (ML) at the interface with the atmosphere.

The specific context of the oceanographic survey constrained the data analysis: biogeochemical responses to a rain event have been scaled in time over a few days (3 - 5), and in space spanning tens of km (40 - 50). Their evolution was restricted to the vertical dimension, integrating lateral exchanges by horizontal diffusion or local advection that occurred at the prescribed space and time scales. In the vertical dimension, exchanges of

nutrients across the ML were controlled by advection due to rapidly changing conditions (MLD fluctuations and nutrient inputs from the atmosphere) rather than diffusion between stationary pools.

In this context, we will i) discuss the classification of stations according to gradients of NO3 concentrations between the ML and the NDLb, ii) discuss the nitrogen budget within the ML

at the short stations considered as 'snapshot', and iii) analyze finely, using a time series of CTD casts, biogeochemical changes within the ML and the NDLb following the atmospheric wet deposition event at FAST site along with the possible modes of transfer of nutrients between these 2 layers.

### 4.1 NO3 dynamics in the ML

Four different groups of stations corresponding to different stages of ML enrichment and relaxation, due to the nutrient inputs from single rain events, have been characterized using the qualitative assessment described in section 2.4, i.e. the differences in NO3 concentration between ML and NDLb. As shown in Fig. 4, this succession of stages is in agreement with the

NO3 fluxes from above and below the ML. Moreover, they provide a temporal scaling of the oceanic response to atmospheric deposition, with a quasi-instantaneous change at the time of the rain event and a 2-days relaxation period to recover pre-event conditions. In terms of spatial scale, although a single rain event was collected on board at ION and FAST sites, the rain radar data indicated the presence of a rain front, including numerous rain events with

abundant rainfalls that occurred in a larger area surrounding the ship location. This happened at ION during the second and third days of sampling (26-27 May), although two short rain events occurred onboard the ship; only the significant rainfall was collected at the end of this



site (29 May). The chemical composition of the rain indicated that the rain sample collected at
ION was under anthropogenic background influence (Fu et al., this issue, in prep. b).

At FAST site a rain occurred aroud the ship on 4 June while rainfall was sampled onboard
only during the night between June 4 and 5. The dust deposition flux estimated from the rain
sampled on board was 12 mg m$^{-2}$ which is among the least intense dust deposition fluxes
recorded in this area from a long time-series of deposition (Vincent et al., 2016). We believe
that the biogeochemical response measured was not only due to a localized event collected on

board but rather was the results of a suite of atmospheric events that had impacted the ML
nutrient contents under the effect of horizontal mixing with a spatial range of the order of the
oceanic mesoscale. This is in agreement with the lagrangian analysis of the surface flow at
FAST (Figs 5 and 12 in Guieu et al. 2020). The evolution of the ML nutrient is indeed
certainly not only linked to the single isolated wet deposition collected onboard but integrates

the total wet atmospheric inputs, spatially patchy over a larger area (~40-50 km radius around
the R/V). However, the chemical characteristics of collected rain probably provide a good
picture of the rain waters composition in the rain fronts.

**4.2 A snapshot of biological fluxes in ML and their link to dry atmospheric deposition**

Enrichment experiments undertaken at the 3 long sampling sites indicated that either single,
or co-limitation by N, P occurred among hprok in the ML (Fig. 6). The dependence of hprok
on nutrients rather than on labile organic carbon sources suggests that they largely compete
with phytoplankton for inorganic nutrients. This is consistent with other studies in the MS
(Van Wambeke et al., 2002, Céa et al, 2017; Sala et al., 2002). Under such conditions of

limitation, it is thus probable, due to the advantage of their small- sized cells and kinetic
systems adapted to low concentration ranges of nutrients (for example for DIP see Talarmin et
al., 2015), that hprok will react rapidly to new phosphorus and nitrogen inputs, like those
coming from the atmosphere. During an artificial *in situ* DIP enrichment in Eastern
Mediterranean, P rapidly circulated through hprok and heterotrophic ciliates, while the

phytoplankton was not directly linked to this 'bypass' process (Thingstad et al., 2005).
Bioassays conducted in the tropical Atlantic Ocean have also shown that hprok respond more
strongly than phytoplankton to Saharan aerosols (Marañón et al. 2010), a pattern that has been
confirmed in a meta-analysis of *in vitro* dust addition experiments (Guieu et al. 2014a).



We considered hprokN demand together with phytoN demand and compared it to
autochthonous (DON breakdown by ectoenzymatic activity) and allochthonous sources
(atmospheric deposition). To our knowledge this is the first time that these fluxes are
compared based on their simultaneous measurements on board. A high variability was
observed among the 10 short stations (Table 2). The regeneration of nitrogen through

aminopeptidase activity was clearly the first provider of N to hprok: 14 to 66% of the hprokN
demand could be satisfied by *in situ* LAP activity. Such percentages may be largely biased by
the conversion factors from C to N and propagation of errors for the LAP hydrolysis rates and
BP rates. However, the C/N ratio of hprok is relatively narrow under large variations of P or
N limitation (6.2 to 8.4; Nagata 1986).

Other regeneration sources exist like direct excretion of NH4 or low molecular weight DON
source with no necessity of hydrolysis prior to uptake (Jumars et al., 1989). For instance,
Feliú et al. (2020) calculated that NH4 and DIP excretion by zooplankton would satisfy 25-
43% and 22-37% of the whole euphotic zone phytoN demand and phytoP demand,
respectively. Such percentages suggest that direct excretion by zooplankton in addition to

ectoenzymatic activity, also substantially provides N for biological activity.

    $N_2$ fixation can also fuel hprok directly as some are heterotrophic diazotrophs (Delmont et al.
2018, and references therein), or indirectly, as part of the fixed $N_2$ rapidly cycles through
hprok (Caffin et al., 2018). Furthermore, a better coupling of N2fix rates with BP rather than
with PP has been observed in the eastern MS (Rahav et al., 2013a). For this reason, we also

examined a hypothetical contribution of N2fix rates to hprokN demand: it was low, ranging
from 1 to 4.5 %. It is consistent with the low N2fix rates observed in the MS (i.e. Rahav et al.,
2013b; Ibello et al., 2010; Ridame et al., 2011; Bonnet et al., 2011), in comparison to oceans
primarily limited by N but not P, such as the south eastern Pacific where N2fix rates are high
(Bonnet et al., 2017) and can represent 3 to 81 % of the hprokN demand (Van Wambeke et

al., 2018).

    Considering LAP as the most relevant regeneration N source, it is clear that even the sum of
LAP activity and $N_2$ fixation were not sufficient to meet hprokN demand (cumulated, those
fluxes contributed 19 to 73% of HbactN demand). With hprok primarily limited by the
availability of N, P or colimited NP, we thus examined possible N sources provided by the

atmosphere (P sources at the short stations are discussed in Pulido-Villena et al, this issue, in
prep.). Dry atmospheric DIN deposition fluxes (29 ± 7 µmol N m$^{-2}$ d$^{-1}$ at the short stations)
were among the lowest recorded for the Mediterranean but were obtained offshore compared



to other measurements carried out at coastal sites where local/regional contamination

contribute significantly to the flux (Desboeufs, in press). On a yearly survey, based on 10

aerosols collectors all around coastal sites of the Mediterranean Sea, dry DIN deposition was

very patchy in space and time with an average of 38 µmol N m$^{-2}$ d$^{-1}$ (Markaki et al, 2010).

Nevertheless, atmospheric deposition fluxes also include organic matter (Djaoudi et al., 2017,

Kanakidou et al., 2018), which is bioavailable for marine hprok (Djaoudi et al., 2020).

Soluble DON from aerosols was unfortunately not determined here. DON solubilized from

aerosols contributes 19 to 42% of the total N dry deposition in MS (Desboeufs, in press;

Galletti et al., 2020). Considering a mean of 32%, DON dry deposition ranged from 8 to 19

µmol N m$^{-2}$ d$^{-1}$ at the short sampling stations. The total N flux provided by the dry

atmospheric deposition (inorg+org) would represent from 14 to 121% of hprokN demand.

Such concomitant measurement confirmed the great spatial variability of the different

contributions. Such percentages can be biased by the assumptions on the deposition velocity

used. We considered averages deposition velocity set at 1 cm sec$^{-1}$ for NO3 and 0.21 cm sec$^{-1}$

for NH4. As NO3 was dominating inorganic dry deposition, it is clear that the choice of 1 cm

sec$^{-1}$ for NO3 influenced largely its contribution. This choice is conditioned by the

predominance of NO3 in the large mode of Mediterranean aerosols as dust or sea salt particles

(e.g. Bardouki et al., 2003). However, the deposition velocity of NO3 between fine and large

particles could range from 0.6 to 2 cm sec$^{-1}$ in the Mediterranean aerosols (e.g. Sandroni et

al., 2007). Even considering the lower value of 0.6 cm sec$^{-1}$ from the literature, the

contribution of atmospheric dry deposition to hprokN demand within the ML would still be

significant, and up to 72%.


We examined if the four groups of stations described in results section 3.1 and based on NO3

disribution were also characterized by particular biological fluxes. Although a large

variability of biological activities were observed in the ML among those 4 groups, some

trends were depicted. LAP activity in ML was higher at the stations of group 1 (with lowest

NO3 concentration in ML and NDLb), in comparison to the group 4 (which exhibited the

larger difference of concentrations between ML and NDLb) (121 ± 54 compared to 46 ± 23

µmol N L$^{-1}$ d$^{-1}$, respectively). The stations from group 1 also presented the lowest DIN dry

deposition flux (26 ± 6 µmol N m$^{-2}$ d$^{-1}$). Finally, it was always in a station from group 4 that

the higher phytoN demand, HbactN demand and N2fix rates were detected. Part of the

variability obtained among the different groups was also related to the instantaneous character





of comparison. Indeed, it is clear that a succession of biogeochemical changes occurs at different time scales during an atmospheric deposition event. Such comparison as a function of time was possible at station FAST and will be discussed below.

**4.3 Biogeochemical response after a wet event – N and P budgets at FAST site**

Rain events are more erratic than dry atmospheric deposition but provide on average much higher nutrient fluxes on an annual basis, e.g. 84% of DIN fluxes in Corsica Island (Desboeufs et al., 2018). At the scale of Mediterranean basin, the annual wet deposition of DIN was found to be 2-8 times higher than that of dry deposition (Markaki et al., 2010). Wet

deposition contributes also significantly to DON atmospheric fluxes in the MS: total (wet + dry) DON atmospheric fluxes ranges between 7 and 367 µmol DON $m^{-2}$ $day^{-1}$ in the NW MS (Frioul Island; Djaoudi et al., 2018) and between 1.5 and 250 µmol DON $m^{-2}$ $day^{-1}$ in the Eastern MS (Lampedusa Island; Galetti et al., 2020), with DON contribution to the total nitrogen flux of 41 ± 14% and 25%, respectively (Djaoudi et al., 2018; Galletti et al., 2020).

In both studies, bulk atmospheric fluxes of DON were positively correlated with precipitation rates, indicating the preponderance of wet deposition over dry deposition.

The coupled atmospheric and biogeochemical water column sampling time series at FAST site was based on casts performed before and after a single rain collected on board on the

5/6/2017 at 3:04, and lasting for 28 min. The maximum net accumulation of NO3 and DIP within the ML after the rainy period reached i) 1520-665=855 µmol N $m^{-2}$ for NO3 (data from FAST+0.24 relative to FAST-0.25 and ii) 283-137=144 µmol P $m^{-2}$ for DIP, (data from FAST+1.05 relative to FAST-0.25). In other terms, based on a mean MLD of 16 m, the rain collected on board would have delivered 0.008 nM DIP and 7 nM NO3 over the whole mixed

layer. The net observed accumulations in the ML (5 nM DIP and 38 nM NO3) are thus much higher than the calculated variation in stocks deduced from the N and P concentrations of this collected rain (Table 4). This is still true when including all P or N chemical forms (dissolved and particulate inorganic + organic). For instance for P, the total atmospheric deposition would increase by ~0.07 nM the P concentration in the ML. One reason for this discrepancy

might be, as developed in section 3.4, that the collected rain sampled on board represented only a fraction of the whole rain episode that affected a large area around the ship. Based on aluminum inventory in the water column (Bressac et al., this issue, in prep.), the total dust deposition having affected the FAST site was 4.6 times higher (55 mg $m^{-2}$) than the dust flux





measured from our single rain event (12 mg m$^{-2}$). It is likely that the deduced new N and P

increases within the ML due to the collected rain are well underestimated. If we consider the
above cited factor of 4.6, the atmospheric flux could have provided an addition of about 33
nM for NO3 and 0.037 nM for DIP (0.35 nM total P) in the ML, more in agreement with the
marine volumetric variation, at least for NO3, and suggesting even that atmospheric nitrate
input could alone explain the increase of NO3 in the ML.


A delay of about 19 h was observed in the maximum net accumulation within the ML
between DIP (FAST+1.05) and NO3 (FAST+0.24). The DIN/DIP ratio in the rain (1438) was
much higher than Redfield ratio. As the biological turnover of DIP in the Mediterranean Sea
is rapid (from min to few h, Talarmin et al., 2015), this indicates that DIP provided by rain

might have behaved differently than DIN. Two different mechanisms can be proposed to
explain the observed delay: (i) processes linked to bypasses and luxury DIP uptake (storage of
surplus P in hprok before a rapid development of grazers (Flaten et al, 2005; Herut et al 2005,
Thingstad et al., 2005) that are later on, responsible for a DIP regeneration process so that
DIP net accumulation is delayed and/or (ii) abiotic processes such as rapid desorption from

large sinking particles followed by adsorption of DIP onto submicronic iron oxides still in
suspension as evidenced experimentally in Louis et al. (2015).
The first proposed mechanism may be supported by the observed increase of BP, along with a
stable PP which suggests an immediate benefit of the new nutrients from rain by hprok rather
than phytoplankton. The so-called luxury DIP uptake is efficient by competing organisms like

hprok (small cells with high surface/volume ratio and DIP kinetic uptake adapted to low
concentrations). It is of course difficult to quantify such variation *in situ* in comparison to
mesocoms/minicosms dust addition experiments, in which clearly heterotrophy is favored
first (Marañón et al., 2010; Guieu et al., 2014b; Gazeau et al., 2020) Few attempts in the field
have been already described, confirming these trends (Herut et al., 2005, Pulido-Villena et al.,

2008) but, as described in the introduction, those studies lacked high frequency sampling.
The second proposed mechanism, the abiotic desorption/adsorption, is compatible with the
observed 19 h delay (Louis et al 2015). Note however that most of the estimates of such
abiotic processes are from dust addition experiments with contrasting results, some showing
this abiotic process of absorption/desorption while the particles are sinking (Louis et al.,

2015), and other not (Carbo et al., 2005) or showing it as negligible in batch experiments
(Ridame et al., 2003). Using natural or artificial dust for enrichment experiments in abiotic



conditions is not simple to realize, or to interpret. It is possible that DIP adsorbed onto large particles rapidly sinks out of the ML, and desorb partly during their transit in the PDLb, where they could stay longer thanks to the pycnoclines barriers.

We tentatively made a P budget: between FAST+1.05 and FAST+2.11 a net decrease of DIP (-87 µmol P m$^{-2}$) was observed in the ML. During that time, advective flux of DIP toward the PDLb was not detectable as DIP concentration within the ML was always lower than within the PDLb (Pulido-Villena et al., this issue, in prep.). This indicated that this DIP was assimilated and/or transformed to DOP via biological processes, and/or adsorbed onto

particles and then exported to PDLb by sedimentation. We integrated PP and BP over that period (34.5 and 19.7 mg C m$^{-2}$, respectively) and, assuming that all the 87 µmol DIP m$^{-2}$ disappearing would be consumed by these organisms, we could estimate a C/P ratio reached in their biomass of ((34.5+19.7)/12*1000)/87=52 which indeed suggests that DIP was not limiting these organisms anymore. Indeed a decrease of C/P quotas may highlight a switch

from P to C limitation for heterotrophic bacteria (Godwin and Cotner, 2015) and from P to N limitation or increased growth rates for phytoplankton (Moreno and Martiny, 2018). Furthermore, as DIP is also recycled via alkaline phosphatase within the ML, we consider also another source of DIP via alkaline phosphatase activity, from which *in situ* activity (see Van Wambeke et al., 2020 for *in situ* estimates) could release 139 µmol DIP m$^{-2}$ during that

period. Assuming also that DIP resulting from of AP hydrolysis was fully assimilated for P biological needs, then C/P ratio would be (((34.5+19.7)/12*1000))/(87+139) =19. This low ratio seems unrealistic for phytoplankton (Moreno and Martigny, 2018) as well as hprok, even growing in surplus C conditions (Makino et al., 2003; Lovdal et al., 2008; Godwin and Cotner, 2015). DIP is abiotically adsorbed on mineral dust particles (Louis et al., 2015), most

of them sinking, and thus constitute a source of export out of the ML. It is also possible, however, that such sorbing process on dust particles allow export of other P-containing organic molecules, as for instance DOP or viruses produced after luxury DIP assimilation. Free viruses, richer in P than N relative to Hprok, could adsorb, like DOM, on dust particles and constitute a P export source. Indeed, free viruses sorb on black carbon particles, possibly

reducing viral infection (Mari et al., 2019; Malits et al., 2015). However, particle quality is a determining factor for DOM or microbial attachment, and what has been shown for black carbon particles is not necessarily true for dust particles. For instance the addition of a Saharan dust to marine coastal waters led to a negligible sorption of viruses to particles and increased abundances of free viruses (Pulido-Villena et al., 2014), possibly linked to an


enhancement of lytic cycles in the ML after relieving limitation (Pradeep Ram and Sime-
        Ngando, 2010).
        We are aware of all the assumptions made here ((i) conversion factors, (ii) *in situ* estimates of
        alkaline phosphatase, (iii) some missing DIP source in the budget, such as the excretion of
        zooplankton estimated to amount 22-37% of the phytoP demand at FAST (Feliú et al., 2020),
iv) lack of knowledge on the different mechanisms linking P to dust particles, and (iv) the
        hypothesis considering the station as a 1D system. Nevertheless, all these results together
        suggest that both luxury consumption by Hprok and export via scavenging on mineral
        particles, likely occurred simultaneously to explain the observed variations of DIP in the ML.

For NO3, and in opposition to the observations for DIP, we observed physical exchanges by
        advection between ML and NDLb. Indeed, NO3 was higher within the ML than within the
        NDLb at FAST+0.24 and FAST+0.53 and then NO3 was lower within the ML than within
        NDLb at FAST+1.05 and FAST+2.11, suggesting a rapid transfer of NO3 in the NDLb. As
        for DIP, we made an N budget within the ML during the period of net NO3 decrease (1520;
1113; 852 and 177 µmol N m$^{-2}$ at FAST+0.24, FAST+0.5, FAST+1.05 and FAST+2.11,
        respectively). This indicates a net loss of 1343 µmol N m$^{-2}$ in 1.8 days within the ML. Over
        these 1.8 days, the time-integrated phytoN and hprokN demands were 682 µmol N m$^{-2}$ and
        378 µmol N m$^{-2}$, respectively, so that total biological demand in ML was 1060 µmol N m$^{-2}$.
        During this period, possible N sources used were stocks of NO3 disappearing (net diminution
assumed to be consumption =1343 µmol N m$^{-2}$), as well as $N_2$ fixation and *in situ*
        aminopeptidase activity which fluxes integrated over that period of time were 13 and 87 µmol
        N m$^{-2}$, respectively, i.e. a total possible source of N amounted 1443 µmol N m$^{-2}$. Keeping in
        mind that the same potential caveats (see above) apply also for the calculation of N budget,
        the biological N demand appeared thus lower than the sources (difference ~380 µmol N m$^{-2}$).
On the other hand at FAST, vertical advective fluxes of NO3 were up to 337 µmol N m$^{-2}$ d$^{-1}$
        (Fig. 4), i.e ~600 µmol N m$^{-2}$ lost from the ML over 1.8 days. From these two different
        approaches, exported NO3 should range between 380 and 600 µmol N m$^{-2}$ during that period.
        We could conclude that about one third of the NO3 accumulated in the ML after the rain was
        exported by vertical advection to the NDLb. This input of new nutrients could benefit
organisms present at the DCM. Indeed, PP and abundances of all 4 phytoplankton groups
        (*Synechococcus*, *Prochlorococcus*, nano and picoeukaryotes) increased at the DCM, 1 or 2
        days after the rain event and PP recovered its initial value before rain (Fig. S3). The increases





in abundances were higher for prokaryotic phytoplankton abundances, as such organisms
would likely benefit from their small size and their ability to use DON/DOP organic
molecules (Yelton et al., 2016).

### 5. Conclusions

This study reports for the first time simultaneous sampling of atmospheric and ocean
biogeochemical parameters to characterize in-situ, in the context of an oceanographic cruise,
the biogeochemical responses to dust deposition within the ML. High-frequency sampling, in
particular at the FAST site, confirmed the transient status of exchanges between the ML and
the NDLb layers. Furthermore, the dust wet deposition event, as usual, was intermittent and
patchy in space and time and we caught only one part of the deposition. We hypothesized a
fully non-stationary and purely vertical evolution of the oceanic response, lasting for a few
days (48 h occupation of the FAST site) and spanning over 40 - 50 km. Under these
assumptions, we assume our sampling representative of a wider-ranging water mass than the
one crossed by the vessel, integrating every local patch of rain event (rather than only the one
received onboard) thanks to horizontal oceanic stirring. Finally, the comparison with the
mesocosms results (where the fertilization is more important due to high dust concentrations)
is hard to extrapolate with the ones presented here.

Nevertheless, our results showed the important role played by the ML in the biogeochemical
and physical processes responsible for transfers of matter and nutrients between the
atmosphere and the nutrient depleted layer below. Thanks to the uses of LWCC technique and
access to nanomolar variations of NO3 and DIP, it was possible to demonstrate the role of the
ML and exchanges of NO3 from the ML to the NDLb by vertical advection when variations
of ML depth occurred simultaneously to transitory accumulation of NO3 after deposition
event. As pictured on Fig. 7, this study confirms previous evidence that hprok in the ML and
the presence of dust particles and mechanism of sinking adsorption/desorption play together a
role in the transfer of P atmospheric deposition from the ML to the nutrient depleted layer.
Our results also show the time sequence occurring after a wet dust deposition event:
accumulation of NO3 in the ML, advection to NDLb, luxury consumption of DIP by hprok
and delayed peaks of DIP, decrease of primary production and its recovery after 2 days
mainly visible in the nutrient depleted layer. The effect of dust deposition is thus a complex
and timely-controlled trophic cascade within the microbial food web. It showed the important



role of intermittent, but strong abiotic effects such as downwelled advective fluxes from the
       ML to the nutrient depleted layers. It will be important to consider these aspects in budgets
       and non stoichiometric models, especially when climate and anthropogenic changes are
       predicted to increase aerosol deposition in the Mediterranean Sea.

**Acknowledgements** :This study is a contribution of the PEACETIME project
       (http://peacetime-project.org), a joint initiative of the MERMEX and ChArMEx components
       supported by CNRS-INSU, IFREMER, CEA, and Météo-France as part of the programme
       MISTRALS coordinated by INSU (doi: 10.17600/17000300). The research of EM was
       funded by the Spanish Ministry of Science, Innovation and Universities through grant
PGC2018-094553B-I00 (POLARIS). The authors thank also many scientists & engineers for
       their assistance with sampling/analyses: Marc Garel, Sophie Guasco and Christian Tamburini
       for AP and LAP activity, Ruth Flerus for TAA, Joris Guittoneau and Sandra Nunige for
       nutrients, Thierry Blasco for POC, Julia Uitz and Céline Dimier for TChl a (analysed at the
       SAPIG HPLC analytical service at the IMEV, Villefranche), Philippe Catala for flow
cytometry analyses, Sylvain Triquet and Franck Fu for atmospheric particulate nitrogen and
       phosphorus.



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



**Table 1.** Main biogeochemical features/trophic conditions during the PEACETIME cruise.
For TYR, ION and FAST sites investigated during several days, means ± sd are indicated.
TChla: Integrated total chlorophyll a (Chla + dvChla). IPP: Integrated particulate primary
production; IBP: integrated heterotrophic prokaryotic production from surface to 200 m
depth.

| | sampling date | Lat. °N | Long. °E | Temp. at 5 m °C | Bottom depth m | DCM depth m | TChl a mg chla m⁻² | IPP mg C m⁻² d⁻¹ | IBP mg C m⁻² d⁻¹ |
|---|---|---|---|---|---|---|---|---|---|
| ST1 | May 12 | 41°53.5 | 6°20 | 15.7 | 1580 | 49 | 35,0 | 284 | 51 |
| ST2 | May 13 | 40°30.36 | 6°43.78 | 17.0 | 2830 | 65 | 32,7 | 148 | 55 |
| ST3 | May 14 | 39°0.8.0 | 7°41.0 | 14.3 | 1404 | 83 | 23,2 | 140 | 77 |
| ST4 | May 15 | 37°59.0 | 7°58.6 | 19.0 | 2770 | 64 | 29,2 | 182 | 66 |
| ST5 | May 16 | 38°57.2 | 11°1.4 | 19.5 | 2366 | 77 | 30,5 | 148 | 51 |
| TYR | May 17-20 | 39°20.4 | 12°35.56 | 19.6 | 3395 | 80 ± 6 | 29 ± 3 | 170 ± 35 | 57 ± 3 |
| ST6 | May 22 | 38°48.47 | 14°29.97 | 20.0 | 2275 | 80 | 18,7 | 142 | 62 |
| ST7 | May 24 | 36°39.5 | 18°09.3 | 20.6 | 3627 | 87 | 24.2 | 158 | 57 |
| ION | May 25-28 | 35°29.1 | 19°47.77 | 20.6 | 3054 | 97 ± 5 | 29 ± 2 | 208 ± 15 | 51 ± 9 |
| ST8 | May 30 | 36°12.6 | 16°37.5 | 20.8 | 3314 | 94 | 31.6 | 206 | 71 |
| ST9 | June 2 | 38°08.1 | 5°50.5 | 21.2 | 2837 | 91 | 36.1 | 214 | 64 |
| FAST | June 2-7 and 9 | 37°56.8 | 2°54.6 | 21.0 | 2775 | 79 ± 8 | 34 ± 8 | 211 ± 57 | 92 ± 11 |
| ST10 | June 8 | 37°27.58 | 1°34.0 | 21.6 | 2770 | 89 | 28,9 | nd | 96 |






**Table 2.** N budget at the short stations within the surface mixed layer (ML). Integrated stocks (NO3, μmol N m$^{-2}$) and fluxes (heterotrophic prokaryotic N demand (hprokN demand), phytoplankton N demand (phytoN demand), in situ leucine aminopeptidase hydrolysis fluxes (LAP), dry atmospheric deposition of NO3 and NH4 (all fluxes in μmol N m$^{-2}$ d$^{-1}$). Values presented as mean ± sd. SD was calculated using propagation of errors of measurements (all data), C/N ratio of 7.3 ± 1.6 (hprokN demand) and 7 ± 1.4 (phytoN demand). MLD: mixed layer depth. na: not available because under LWCC detection limits.

| | | stocks | biological fluxes | | | | Dry deposition | |
|---|---|---|---|---|---|---|---|---|
| stations | MLD | NO3 | phytoN demand | hprokN demand | LAP | N$_2$ fixation | NO3 | NH4 |
| | m | μmol N m$^{-2}$ | μmol N m$^{-2}$ d$^{-1}$ | μmol N m$^{-2}$ d$^{-1}$ | μmol N m$^{-2}$ d$^{-1}$ | μmol N m$^{-2}$ d$^{-1}$ | μmol N m$^{-2}$ d$^{-1}$ | μmol N m$^{-2}$ d$^{-1}$ |
| ST1 | 21 | na | 1468 ± 325 | 184 ± 40 | 121 ± 28 | 14.6 ± 1.5 | 18.6 ± 1.4 | 1.5 ± 0.3 |
| ST2 | 21 | na | 481 ± 161 | 163 ± 35 | 48 ± 24 | 10.7 ± 1.1 | 23.7 ± 2.2 | 4.1 ± 0.9 |
| ST3 | 11 | na | 282 ± 82 | 126 ± 28 | 40 ± 17 | 7.8 ± 0.8 | 33.8 ± 3.6 | 4.7 ± 0.5 |
| ST4 | 15 | na | 246 ± 80 | 132 ± 28 | 83 ± 20 | 10.7 ± 1.1 | 23.8 ± 2.9 | 6.3 ± 2.6 |
| ST5 | 9 | 261 ± 22 | 112 ± 29 | 42 ± 9 | 17 ± 12 | 4.8 ± 0.5 | 27.0 ± 7.5 | 7.9 ± 1.8 |
| ST6 | 18 | 162 ± 14 | 410 ± 116 | 204 ± 44 | 48 ± 24 | 9.1 ± 0.9 | 15.0 ± 0.6 | 9.3 ± 0.7 |
| ST7 | 18 | 162 ± 14 | 226 ± 123 | 148 ± 33 | 83 ± 18 | 10.5 ± 1.1 | 23.6 ± 1.9 | 8.0 ± 1.2 |
| ST8 | 14 | 911 ± 77 | 274 ± 66 | 130 ± 33 | 25 ± 8 | 4.3 ± 0.5 | 13.4 ± 1.7 | 3.8 ± 0.6 |
| ST9 | 7 | 819 ± 70 | 259 ± 70 | 85 ± 22 | 21 ± 6 | 3.4 ± 0.4 | 27.4 ± 3.8 | 13.5 ± 0.8 |
| ST10 | 19 | 2074 ± 176 | 495 ± 31 | 294 ± 64 | 42 ± 26 | 13.6 ± 1.4 | 23.9 ± 3.4 | 4.1 ± 0.4 |






**Table 3**. Characteristics of the 2 rains collected during the PEACETIME cruise. The collector surface was 0.045 m$^2$

| event | Rain 1 | Rain 2 |
|---|---|---|
| Location | ION | FAST |
| Date and local time | 29/05 05:08-6:00 | 05/06 02:36-3:04 |
| Duration min | 52 | 28 |
| Collected volume ml | 118 | 87.9 |
| DIP Flux nmol P m$^{-2}$ | 495.5 | 129 |
| DOP Flux nmol P m$^{-2}$ | 728.2 | 103 |
| POP fluxes nmol P m$^{-2}$ | 179 | 994 |
| NO3 Flux µmol N m$^{-2}$ | 50.0 | 115.7 |
| NH4 Flux µmol N m$^{-2}$ | 53.2 | 70.6 |
| DIN Flux µmol N m$^{-2}$ | 103 | 187 |






**Figure legends**

**Figure 1.** Nitrate (NO3) aerosol concentration along the PEACETIME transect. The location of two rain events are indicated by large black circles. Stations ST 1 to 4 were not sampled for analysis of nutrients at nanomolar level.

**Figure 2.** Representation of the mixed layer (ML), the bottom of the nitrate (NO3) depleted layer (NDLb), delineated by the nitracline depth and the mixed layer depth (MLD).

**Figure 3**. a) Evolution of the wind speed during the cruise PEACETIME. The stations are indicated in yellow and dates in black. Vertical dotted lines delineate the beginning and the end of occupation of the ship at TYR, ION and FAST sites. The two rain events collected on
board are indicated in solid vertical red arrows and surrounding observed rains by horizontal dashed red arrows. b) 0-100 m vertical distribution of nitrate (NO3) with depth. The MLD (in red) and nitracline (in brown) are indicated.

**Figure 4**. Average concentration of nitrate (NO3) in the ML and the NDLb, and NO3 flux from ML to NDLb. The stations have been classified into 4 types (1 in blue, 2 in green, 3 in
yellow, 4 in red, see section Results and Table S1 for definitions). Error bars are indicated by standard deviation around average values for nitrate concentrations, and error propagation for the flux from ML to NDLb using a 0.5 m uncertainty in the MLD variation.

**Figure 5.** Evolution within the ML of heterotrophic prokaryotic production (BP), particulate primary production (PP), heterotrophic prokaryotes (hprok) and *Synechococcus* (syn)
abundances at the station FAST. The mixed layer depth is indicated in red line.

**Figure 6**. Results of the enrichment experiment. BP reached after 36 h of enrichment in the dark. C: control unamended, P:+DIP, N:+NO3+NH4, G:+glucose, and combinations of these elements in PN, NG, PG and NPG. Stars show significantly different BP compared to the
unamended control .

**Figure 7**. Synthetic view of biogeochemical processes and exchanges between ML and NDLb before the rain and evolution after the rain.



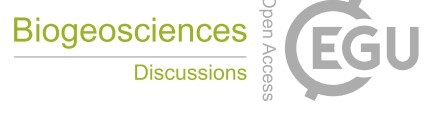

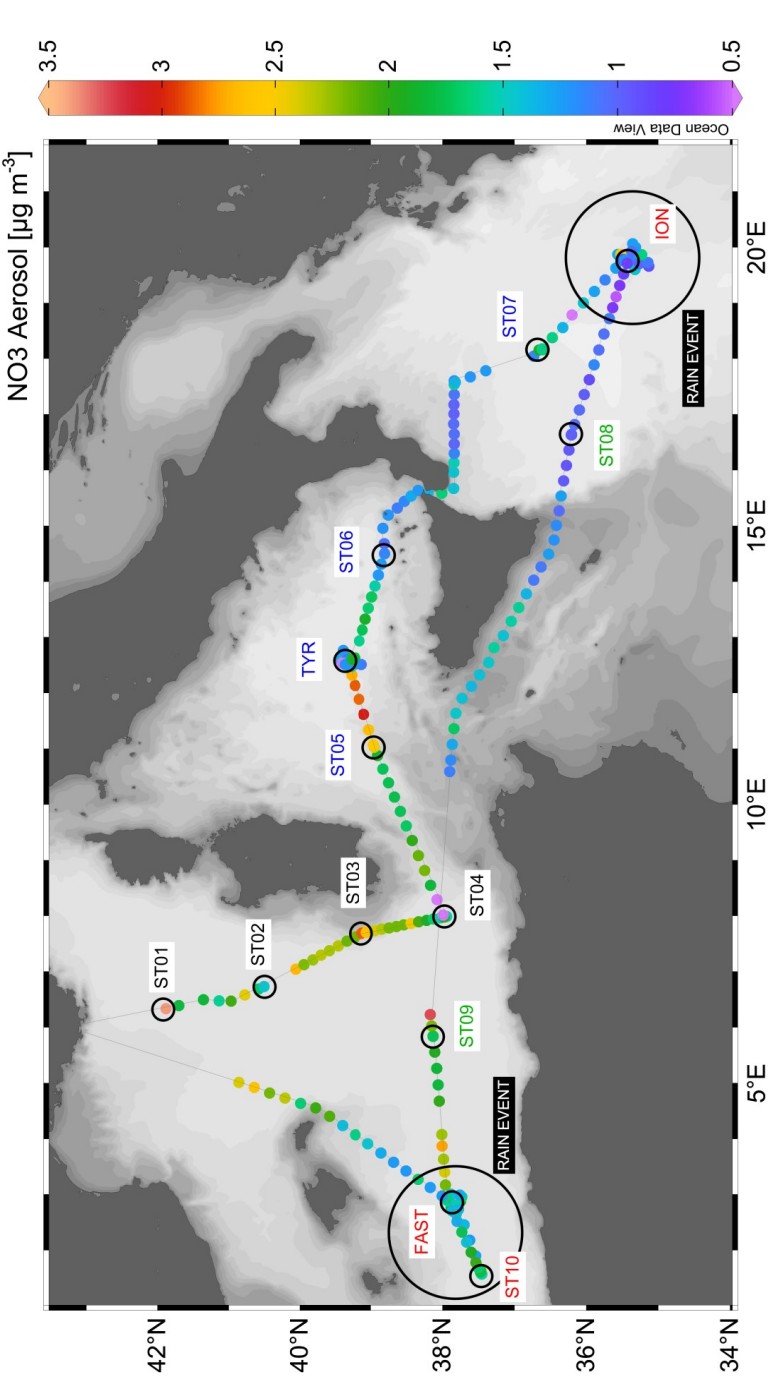

Fig. 1



Fig.2

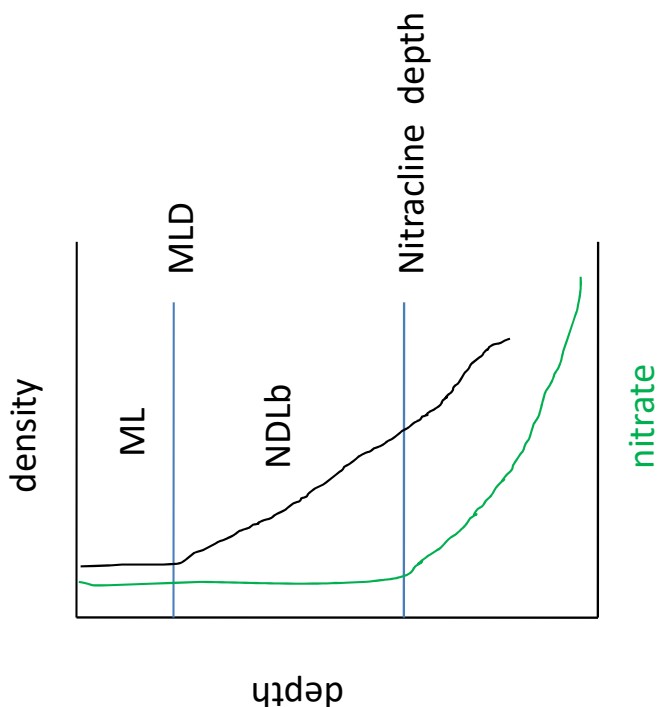



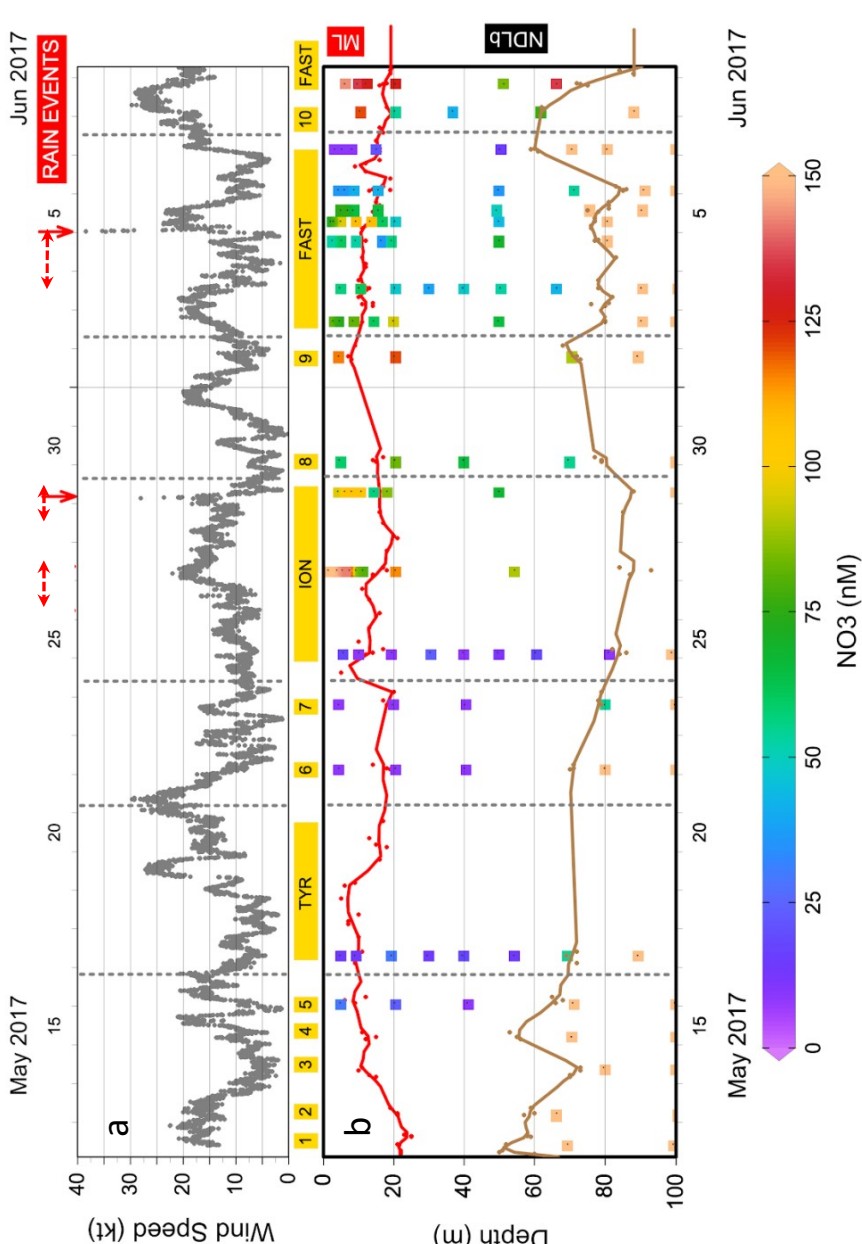

Fig. 3



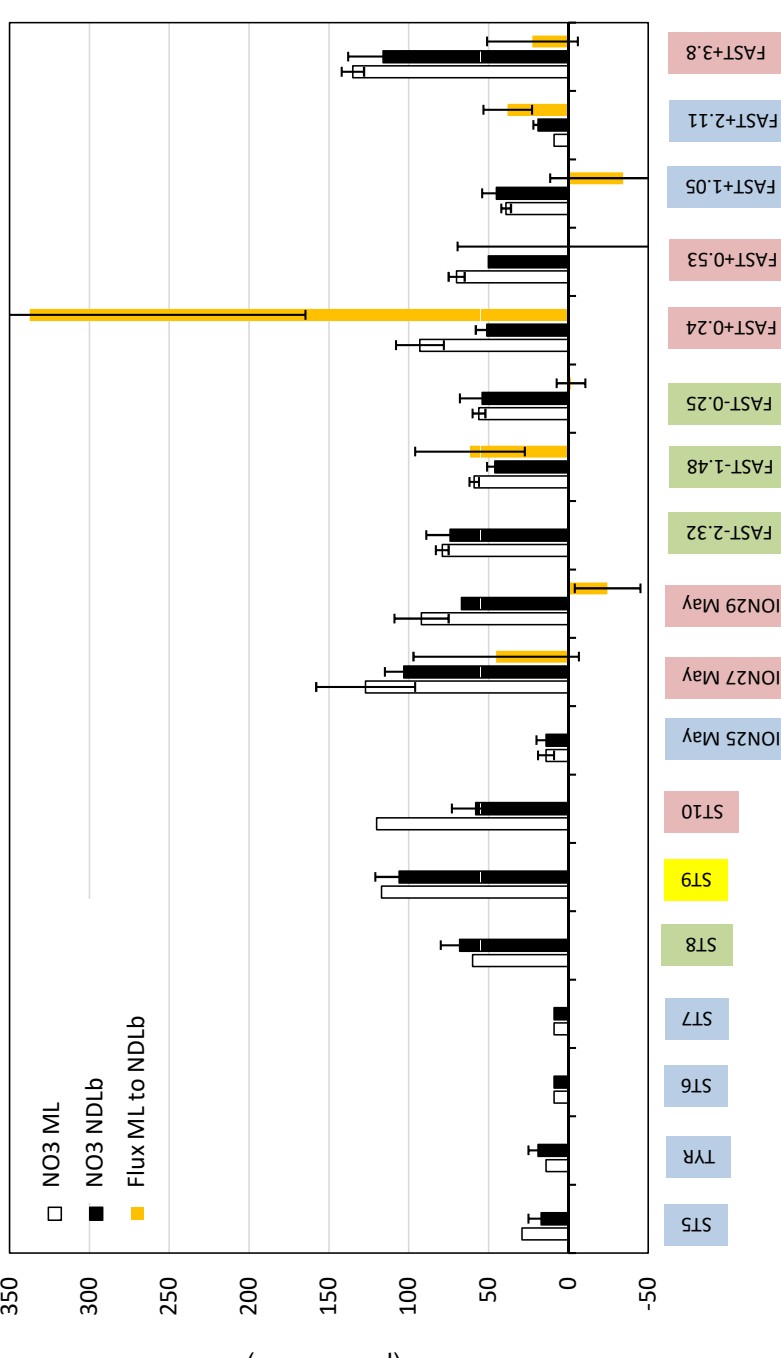

Fig. 4



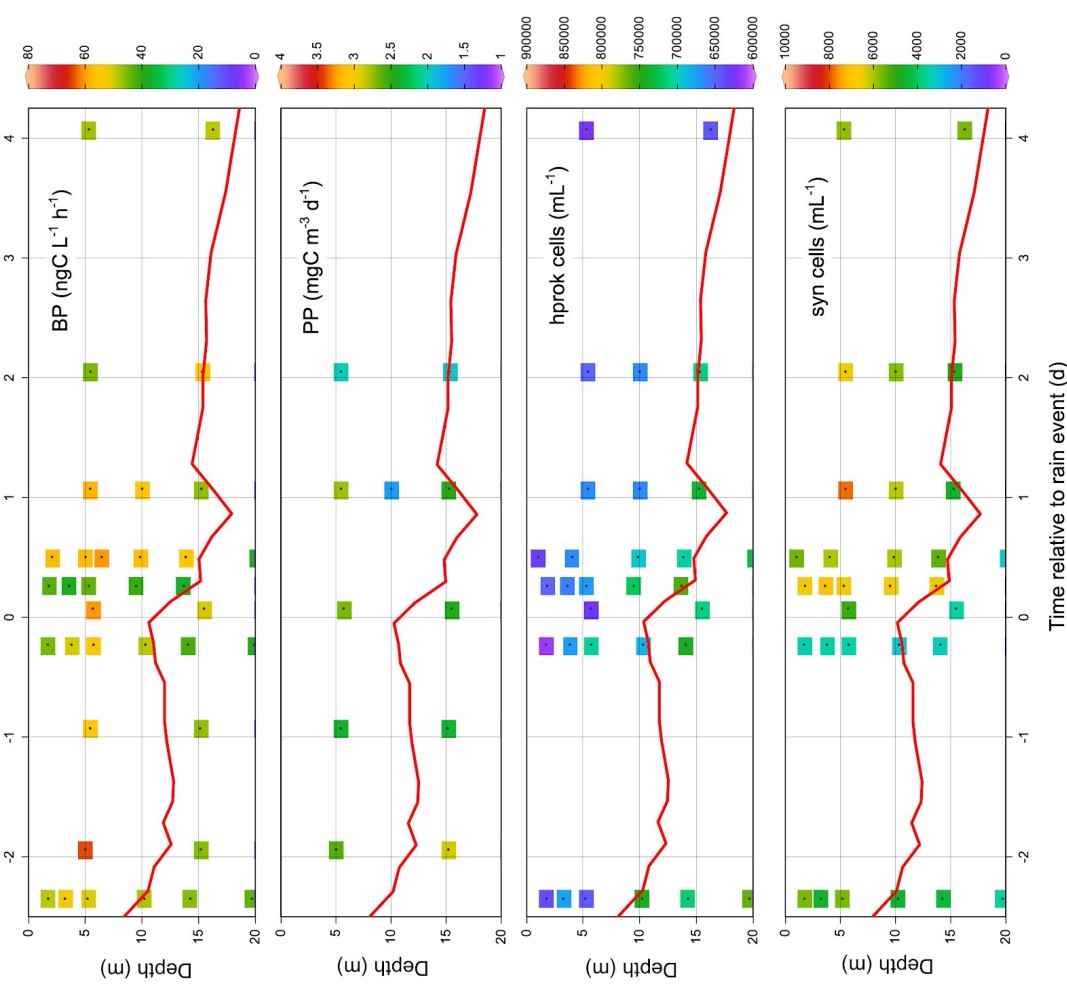

Fig. 5



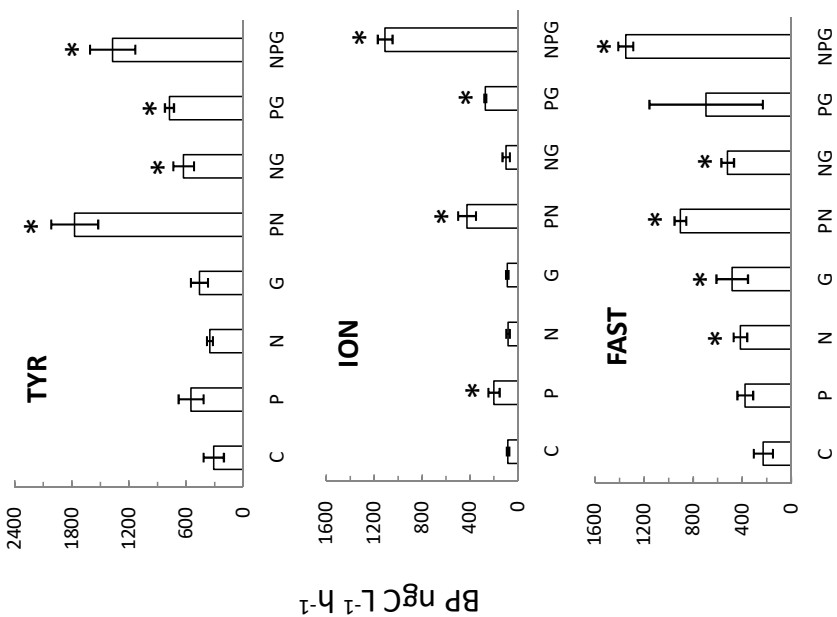

Fig. 6



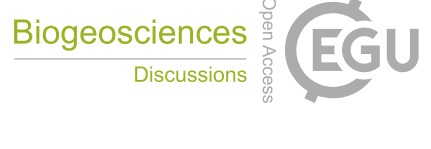

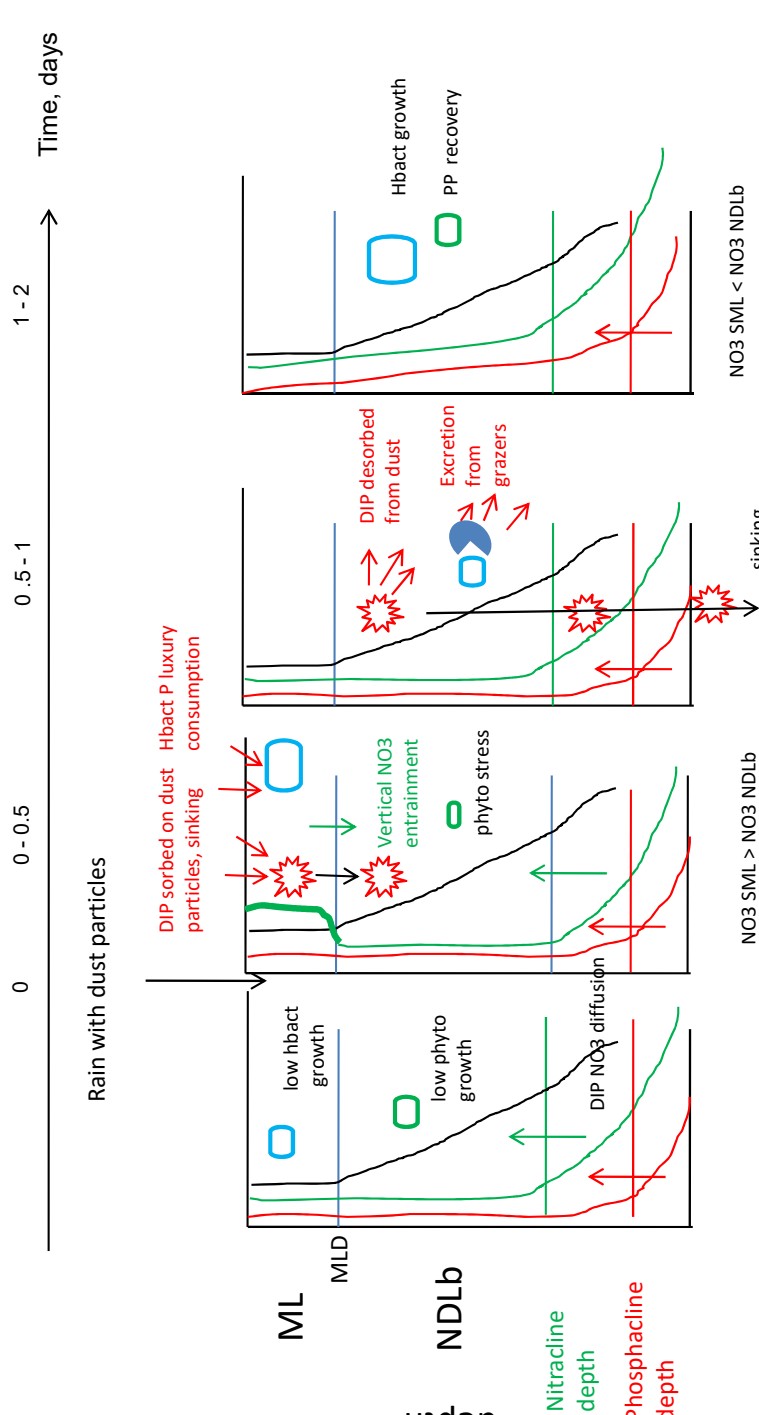

Fig. 7