# Peer review of "Influence of atmospheric deposition on biogeochemical cycles in an oligotrophic ocean system"

_Biogeosciences, 2020_

## Referee Comment (RC1) · Maurizio Ribera d'Alcala' (Referee) · 31 Jan 2021

I want to apologize with the Editor and the Authors for the delay in sending this report, due to family reasons overlapped with the unusual period we are going through because of the pandemics.

Referee report for the manuscript bg-2020-411: **Influence of atmospheric deposition on biogeochemical cycles in an oligotrophic ocean system** by *France Van Wambeke, Vincent Taillandier, Karine Desboeufs, Elvira Pulido-Villena, Julie Dinasquet, Anja Engel, Emilio Marañón, Céline Ridame, Cécile Guieu*

This paper is clearly one among several contributions for a special issue of BG illustrating and discussing the results of the basin scale experiment PEACETIME, carried out during summer 2017 in the Mediterranean sea to assess the impact of atmospheric depositions on the functioning of plankton food web in the basin. Therefore it describes a small set among the numerous processes that have been studied during the cruise, namely, the weighted support to phytoplankton and heterotrophic bacteria production and nutrient assimilation in the surface layer by different forms of bioavailable Nitrogen (N) and Phosphorus (P). The obvious focus is on the forms derived by the atmospheric deposition. This is just a piece of a mosaic whose whole picture, I assume, will be published either in this special issue or somewhere else. The key question behind the experiment was the extent to which and by which mechanisms atmospheric deposition modulates biogeochemical processes in the basin. This is also reflected in the title and the manuscript addresses a key phase: the first processing of new nutrients in the surface layer. Therefore the results represent the short time response, even though they likely embed also responses due to processes occurring over some time before the event (see below). To better dissect the processes in the surface layer the authors divide it in two sublayers, the Mixed Layer (ML) proper which is the layer directly affected by air sea interactions and, therefeore, the entry point of atmospheric inputs, and a second layer spanning from the ML Depth and the depth of the nutriclines (N or P) identified as the depth of the 'heaviest' isopycnal with Dissolved Inorganic Nitrogen (DIN) or Phosphorus (DIP) values equal to zero. The latter determined by extrapolation in DiX-density diagrams. Considering what is reported in Table S1 I assume that the zero value is obtained plotting the concentration determined with the segmented flow analyzer (a clarification on line 333 would help). This allowed them to discriminate among the different sources or sinks of surface DiX to better identify the weight of the atmospheric component. Different N and P chemical forms where analyzed and linked to the different sources and sinks, namely DIN derived from N-fixation, DIN from atmospheric inputs, in situ Total Dissolved Nitrogen (TDN) and Total Hydrolizable Aminoacids (TAA) for Nitrogen and DIP from atmospheric inputs, total Dissolve Organic Phosphorus (DOP), and its labile component. Three stations were sampled for longer times, namely those where wet deposition events were occurring or had occurred just before sampling, which allowed, with the caveat of advective processes going on, of better reconstructing the vertical dynamics of N and P in relation to pico-plankton, the dominant component in the sampled area, activity. The main oucome of the paper is a quantification of the fraction of N-demand and P demand by pico-planktonic autotrophs and heterotrophs supplied by the atmospheric inputs as compared with the other sources of recycled or existing forms. An additional outcome is the difference in the N and P dynamics in the surface layer after input which may hint to a differential response of the picoplankton community to the supply of the two elements as weel as to particle scavenging.

The paper contains many useful data and tries to condense a big experimental effort. The need to present them in full, makes the text a bit heavy to read (e.g., section 3.1). In the discussion the authors make the commendable effort to examine all the possible processes, already reported in the literature, which may explain the patterns they observed but this, at the end, does not provide the reader with an answer, or at least the answer preferred by the authors. The fact that in such large experiments with so many scientists involved, there is a need to divide the informations in many papers weaken a bit some of them because many questions that arise reading one are likely answered in other papers of the same

issue. Of course I support the publication of the paper for the valuable information it provides, for the fact that is one piece of a large picture, and because the methods used are robust.

My suggestion to make the paper more punchy are the following:

1. Let the tables to summarize the data and shorten the description highlighting the most significant result. The text is definitely long.

2. Do not include the enrichment experiments. They do not add too much unless you discuss them integrating them with the other information.

3. Discuss what are your conclusions about the significant variability among the different sites having in view the possoble impact at basin scale. The present conclusions discuss other aspects.

Additional comments

l.215-218 *"The concentrations used were averages from PILS-IC analysis obtained during the occupation of each short station, and averages between two successive casts during site occupations, except for ST1 where concentrations were issued from filter by IC analyses after water extraction."* It is not clear to me why for dry deposition it was taken a quasi-instantaneous value instead of an estimate of the deposition during a few hours before the sampling. It would be intersting to mention here the variability of dry deposition along the cruise track which, should be part of the Fu et al. (in preaparation) paper and to discuss how representative are quasi-instantaneous values. Likewise for filter samples. This aspect is indeed considered only for the stations that were sample for many days because of wet deposition events (see l.628-632).  I am also wondering, not being an expert, if the ions solubilization efficiency and time of aerosol particles in PILS is the same than that of sea water.

l.258 space after (...Dickson)

l.345 For what I understood the advective flux is an entrainment/detrainment mechanism sensu *Cullen, J. J., Franks, P. J., Karl, D. M., & Longhurst, A. L. A. N. (2002). Physical influences on marine ecosystem dynamics. The sea, 12, 297-336*. If true, an additional sentence may better clarify which processes the authors refer to.

l.441-443 It is a little confusing putting together the rates of opposite processes, the demand, which is the theoretical intake given certain compositional ratios, and the increase in availability. Since the sentence that follows clarifies the point I would remove the part of the sentence
realated to the release rate or I would write "...   *(hprokN demand), which is confronted by an in situ LAP…)*

l.629 this phrase may be misleading. Indeed what is measured is not the integral but the time-space weighted average of the fluxes

l.447 correct *dominatedby*

l.739 and the following. Please rephrase, since you do not *deliver* neither *accumulate* concentrations, as it is more properly written on l.744.

References

Pulido-Villena et al (2019) is cited on l.243 but is not included in the refs.

The following papers are in the References but are never cited in the text

Holmes, R. M., Aminot, A., Kérouel, R., Hooker, B. A,.and Peterson, B. J.: A simple and precise method for measuring ammonium in marine and freshwater ecosystems, Can. J. Fish. Aqua. Sci., 56, 1801-1808, 1999.

Karl, D. M.: Microbially mediated transformations of phosphorus in the sea: New views of an old cycle. Ann. Rev. Mar. Sci. 6: 279–337, doi: 10.1146/annurev-marine-010213-135046, 2014.

Krom, M. D., Herut, B., and Mantoura, R. F. C.: Nutrient budget for the eastern Mediterranean: Implication for phosphorus limitation, Limnol. Oceanogr, 49, 1582-1592, doi: 10.4319/lo.2004.49.5.1582, 2004.

Krom, M. D., Emeis, K.-C., and Van Capellen, P.: Why is the eastern Mediterranean phosphorus limited? Progress In Oceanography, 85, 236-244, 2010.

Lindroth, P., and Mopper, K.: High performance liquid chromatographic determination of sub picomole amounts of amino acids by precolumn fluorescence derivatization with o-phthaldialdehyde, Anal. Chem. 51, 1667–1674, 1979.

Thingstad, T. F., and Rassoulzadegan, F.: Nutrient limitations, microbial food webs, and 'biological C-pumps': suggested interactions in a P-limited Mediterranean, Mar. Ecol. Prog. Ser., 117, 299-306, 1995.

Zhang, J.-Z., and Chi, J. : Automated analysis of nano-molar concentrations of phosphate in natural waters with liquid waveguide, Environ. Sci. Technol., 36, 1048-1053, doi : 10.1021/es011094v, 2002.

---

## Referee Comment (RC2) · Anonymous Referee #2 · 12 Mar 2021

Comments on the manuscript bg-2020-411: Influence of atmospheric deposition on biogeochemical cycles in an oligotrophic ocean system

This manuscript by Van Wambeke et al. aims at investigating the impact of atmospheric depositions on the biogeochemical properties and processes in the Mediterranean Sea surface mixed layer. For this purpose, the authors present a large amount of data, collected both in the water column and in the atmospheric depositions, along a transect covering both the Western and Eastern basins of the Mediterranean Sea, during the PEACETIME cruise. The authors also presented the results of an enrichment experiment. The strength of this paper, as the authors stated in their conclusions, is that

it provides simultaneous sampling on both atmospheric depositions and the surface ocean on a large portion of the Mediterranean Sea. Moreover, as the authors highlighted in their introduction, the "in situ" study of atmospheric depositions is very rare due to obvious limitations. For these reasons, I think these data should be published. However, the manuscript still needs a bit of refining. I hope that the following comments will be helpful to the authors.

Main considerations: 1) I understand the difficulty of writing a paper with so many results. However, an effort need to be made to shorten the whole manuscript. In the current form I find it quite hard to read, it is too long, one can get lost while reading it. I suggest the authors try to smooth it and make it shorter. There are long sentences that can be shortened, or written with fewer words.

2) I think that the setup of the enrichment experiment is somehow in contrast with the goal of this paper, which is studying the "in situ" effect of atmospheric deposition. As the authors highlighted in the introduction, these experiments are simplifying the natural system. How can we relate the results of an experiment carried out into a 60 ml bottle with what happens in the natural environment? The authors themselves conclude that the results of the experiment cannot be compared with the "in situ" observations. I suggest removing this part.

3) It is clear that a lot of the results obtained from this cruise/project are presented in other papers that are currently under review in this issue or are being prepared. There is a bit of confusion about some data, reported as results in this study, but at the same time citing other papers (under review or in preparation). In particular, I refer to the following: Lines 418-421, DIP results in Pulido-Villena et al. Lines 422, PP and BP in Maranon et al., 2020 Lines 432-435, LAP results in Van Wambeke et al., 2020 Lines 515-517, Citing the results from Fu et al. The authors should clarify, if the results presented are already been present in the cited papers, they should be considered in the discussion section and not presented as results (and therefore also removed from the methods section). If this is the case, it would also help to make the whole

manuscript shorter and more readable.

4) There are way too many citations of articles that are "in preparation", I have counted at least 17. Citing a paper that is in preparation is usually not recommended (sometimes even not allowed), the data are not available and there is no guarantee that they will be. These citations need to be strongly reduced.

Minor comments:

Lines 169-172: This information is contained in table 1, they can be removed from the text and cite the table. Please also check that for stations ION and TYR the dates in the methods and table 1 do not correspond

Lines 196 – 198: Define high and low frequency

Lines 405-416: This division in groups could be summarized in a table, to be more clear to the reader

Lines 425 – 427: Integrated PP and BP, how were they calculated? This information is missing in the methods

Paragraph 3.4 is a mixture of results and discussion and extra information that go beyond the results

---

## Author Comment (AC1) · 9 Apr 2021

**Comments by Maurizio Ribera d'Alcala and our responses**

Referee report for the manuscript bg-2020-411: **Influence of atmospheric deposition on biogeochemical cycles in an oligotrophic ocean system** by *France Van Wambeke, Vincent Taillandier, Karine Desboeufs, Elvira Pulido-Villena, Julie Dinasquet, Anja Engel, Emilio Marañón, Céline Ridame, Cécile Guieu*

This paper is clearly one among several contributions for a special issue of BG illustrating and discussing the results of the basin scale experiment PEACETIME, carried out during summer 2017 in the Mediterranean sea to assess the impact of atmospheric depositions on the functioning of plankton food web in the basin. Therefore it describes a small set among the numerous processes that have been studied during the cruise, namely, the weighted support to phytoplankton and heterotrophic bacteria production and nutrient assimilation in the surface layer by different forms of bioavailable Nitrogen (N) and Phosphorus (P).

The obvious focus is on the forms derived by the atmospheric deposition. This is just a piece of a mosaic whose whole picture, I assume, will be published either in this special issue or somewhere else. The key question behind the experiment was the extent to which and by which mechanisms atmospheric deposition modulates biogeochemical processes in the basin. This is also reflected in the title and the manuscript addresses a key phase: the first processing of new nutrients in the surface layer. Therefore the results represent the short time response, even though they likely embed also responses due to processes occurring over some time before the event (see below).

To better dissect the processes in the surface layer the authors divide it in two sublayers, the Mixed Layer (ML) proper which is the layer directly affected by air sea interactions and, therefore, the entry point of atmospheric inputs, and a second layer spanning from the ML Depth and the depth of the nutriclines (N or P) identified as the depth of the 'heaviest' isopycnal with Dissolved Inorganic Nitrogen (DIN) or Phosphorus (DIP) values equal to zero. The latter determined by extrapolation in DiX-density diagrams. Considering what is reported in Table S1 I assume that the zero value is obtained plotting the concentration determined with the segmented flow analyzer (a clarification on line 333 would help).

This allowed them to discriminate among the different sources or sinks of surface DiX to better identify the weight of the atmospheric component. Different N and P chemical forms where analyzed and linked to the different sources and sinks, namely DIN derived from N-fixation, DIN from atmospheric inputs, in situ Total Dissolved Nitrogen (TDN) and Total Hydrolizable Aminoacids (TAA) for Nitrogen and DIP from atmospheric inputs, total Dissolve Organic Phosphorus (DOP), and its labile component.

Three stations were sampled for longer times, namely those where wet deposition events were occurring or had occurred just before sampling, which allowed, with the caveat of advective processes going on, of better reconstructing the vertical dynamics of N and P in relation to pico-plankton, the dominant component in the sampled area, activity. The main outcome of the paper is a quantification of the fraction of N-demand and P demand by pico-planktonic autotrophs and heterotrophs supplied by the atmospheric inputs as compared with the other sources of recycled

or existing forms. An additional outcome is the difference in the N and P dynamics in the surface layer after input which may hint to a differential response of the picoplankton community to the supply of the two elements as well as to particle scavenging.

The paper contains many useful data and tries to condense a big experimental effort. The need to present them in full, makes the text a bit heavy to read (e.g., section 3.1). In the discussion the authors make the commendable effort to examine all the possible processes, already reported in the literature, which may explain the patterns they observed but this, at the end, does not provide the reader with an answer, or at least the answer preferred by the authors.

The fact that in such large experiments with so many scientists involved, there is a need to divide the informations in many papers weaken a bit some of them because many questions that arise reading one are likely answered in other papers of the same fact that is one piece of a large picture, and because the methods used are robust.

We thank the referee for his constructive comments. To summarize changes made on the revised version: we moved some of the information in the supplement section (the enrichment experiment), we added a new table (S2), and we shortened the text and reorganized some of the discussion. See all our response in blue and citations of parts of the revised version in italics. A .doc version with tracking changes will help to visualize all the changes provided.

My suggestion to make the paper more punchy are the following:

1. Let the tables to summarize the data and shorten the description highlighting the most significant result. The text is definitely long.
As far as possible, unnecessary citation of numbers when the data are also reported on tables were removed.
The M&M, results and figures referring to the enrichment experiment have been moved in the supplement. In the supplement section, we also added a table (Table S2) reporting the stocks and fluxes at ION and FAST Stations. Section 3.3 has been reorganized in a more logical order; section 3.4 has been shortened, as well as the discussion. All in all, the ms is now reduced by a hundred lines (~2 pages).

2. Do not include the enrichment experiments. They do not add too much unless you discuss them integrating them with the other information.
The methodology and result description of the enrichment experiments have been moved to the supplementary section, as well as Fig 6.

3. Discuss what are your conclusions about the significant variability among the different sites having in view the possible impact at basin scale. The present conclusions discuss other aspects.
We think that a snapshot of 10 short stations at a given single period of the year is not enough information to discuss inter-basin variability or impact at the basin scale. Nevertheless, we added a sentence in the discussion showing that the longitudinal dry atmospheric deposition was less variable than biological fluxes within the ML.
We restricted the discussion (which is already very dense) to the contributions of the dry atmospheric flux to the biological needs of phytoplankton and heterotrophic bacteria, as well as on the short term effects of rain, which could have been observed in any part of the basins.

Additional comments
l.215-218 "*The concentrations used were averages from PILS-IC analysis obtained during the occupation of each short station, and averages between two successive casts during site occupations, except for ST1 where concentrations were issued from filter by IC analyses after water extraction.*"
It is not clear to me why for dry deposition it was taken a quasi-instantaneous value instead of an estimate of the deposition during a few hours before the sampling.
PILS-IC measurements have a time-resolution of 70 min. for NO3 analysis and ~30 min. for NH4 analysis. The ship occupied each short station during a few hours (on average 0.33 ± 0.13 days). In order to compare atmospheric deposition and bulk in situ nutrient stocks in the water column at the same time but also at the same place, we didn't use the instantaneous measurements before cast for estimating dry deposition, but we considered the average PILS data acquired during the occupation of the short stations. For example, for station 5 from 16/05 at 4:00 to 16/05 at 11:00, we averaged the PILS data for this period, that comprises 6 measurements for NO3 and 11 measurements for NH4. For long stations, TYR, ION and FAST, we integrated PILS data from the arrival at the station and the first cast, then over the time interval between each cast (that comprises on average about ~15 measurements for NO3 and ~30 for NH4).

We modified the M&M section to make this point clearer.

It would be interesting to mention here the variability of dry deposition along the cruise track which, should be part of the Fu et al. (in preparation) paper and to discuss how representative are quasi-instantaneous values. Likewise for filter samples. This aspect is indeed considered only for the stations that were sample for many days because of wet deposition events (see l.628-632)
The concentrations and fluxes from dry deposition (DIP on filters, NO3 and NH4 from PILS-IC) are not instantaneous but integrate measurements over the few hours of occupation of the short duration station. DIP is derived from one filter sampling aerosols during the whole occupation of the short duration station and consequently with no possible access to a shorter time-variability. In contrast, as the PILS-IC technique allows to determine NO3 and NH4 in aerosols every 70 and 30 minutes respectively, for these parameters the short term time-variability is accessible. It is quantified in the sd cited on Table 2, which is calculated from the variability of the NO3 and NH4 concentrations solubilized from aerosols during the occupation of the short duration stations. This information is now detailed in the legend of Table 2.

The variability of NO3 concentration solubilized from aerosols is also visible on Figure 1.

I am also wondering, not being an expert, if the ions solubilization efficiency and time of aerosol particles in PILS is the same than that of sea water.
PILS-IC is one of the most used device for online measurement of water-soluble inorganic ions in atmospheric aerosol particles. PILS-IC is a method which is optimised to extract water-soluble fraction of N species in aerosols (water being ultrapure water), whatever the type of particles (Orsini et al., 2003). Dissolution experiments from atmospheric aerosols show that the solubilities of nitrate and ammonium are similar in seawater or pure water, i.e. 100% (e.g. Chen et al., 2006). Under these conditions, we considered that the NO3 and NH4 values obtained by aerosols solubilisation by PILS-IC are comparable to the DIN values brought by the dry deposition of these aerosols in sea water.

Orsini, D. A., Ma, Y., Sullivan, A., Sierau, B., Baumann, K., and Weber, R. J., Refinements to the Particle-Into-Liquid Sampler for Ground and Airborne Measurements of Water Soluble Aerosol Compositions. Atmos. Environ., 37:1243–1259, 2003.

Chen, Y., J. Street, A. Paytan, Comparison between pure-water and seawater soluble nutrient concentrations of aerosols from the Gulf of Aqaba, Mar. Chem., 101, 141-152, 2006.

l.258 space after (...Dickson)
done

Line 333 : Considering what is reported in Table S1 I assume that the zero value is obtained plotting the concentration determined with the segmented flow analyzer (a clarification on line 333 would help).
Yes it is, we wrote 'micromolar' concentrations

l.345 For what I understood the advective flux is an entrainment/detrainment mechanism sensu *Cullen, J. J., Franks, P. J., Karl, D. M., & Longhurst, A. L. A. N. (2002). Physical influences on marine ecosystem dynamics. The sea, 12, 297-336*. If true, an additional sentence may better clarify which processes the authors refer to.
Right, the term 'advection' has been used as complementary to 'diffusion', however the advective flux is associated to entrainment/detrainment *sensu* Cullen et al. (2002). The following sentence has been added to clarify the underlying mechanism of vertical advection, in the revised version:
*'The advective flux $F_{ADV}$ corresponds either to the entrainment of deeper water in the mixed layer by erosion of the near-surface pycnocline, or to the detrainment of waters below the mixed layer by restratification, depending on the variations of wind stress and solar heating (Cullen et al., 2002). It is expressed by the variation ...'*

The reference Cullen et al. (2002) has been added in the revised manuscript.

l.441-443 It is a little confusing putting together the rates of opposite processes, the demand, which is the theoretical intake given certain compositional ratios, and the increase in availability. Since the sentence that follows clarifies the point I would remove the part of the sentence realated to the release rate or I would write "... *(hprokN demand), which is confronted by an in situ LAP...)*
done

l.629 this phrase may be misleading. Indeed what is measured is not the integral but the time-space weighted average of the fluxes.
All this part has been modified in the revised version as we finally calculated DIN and P (PP, DOP, DIP) wet deposition events by multiplying concentration calculated from analysis of collected rain by total precipitation estimated from radar data:
The M&M section 2.2.1 was modified as: *'In the rains collected onboard the ship, NO3, NH4 and dissolved inorganic phosphorus (DIP) were also determined by ion chromatography after recovery of the dissolved fraction in the samples. Then, the wet deposition fluxes of these nutrients were estimated from the measured concentrations in the dissolved fractions of rains,*

*multiplied by the total precipitation. This total precipitation was issued from the hourly total precipitation accumulated during the rain event over the region from ERA5 hourly data reanalysis (Hersbach et al., 2018). Total precipitation was obtained by adding hourly rainfall on the grid-points (0.25° x 0.25°) spanning the ship location, more or less 1° around this central grid-point for integrating the regional variability.'*

All the discussion section (the beginning of 3.4 section) reporting risk of underestimation of rain fluxes was consequently removed. Note that the updated rain fluxes considering the total precipitation from radar data do not change the main conclusions of our ms.

Hersbach, H., Bell, B., Berrisford, P., Biavati, G., Horányi, A., Muñoz Sabater, J., Nicolas, J., Peubey, C., Radu, R., Rozum, I., Schepers, D., Simmons, A., Soci, C., Dee, D., Thépaut, J-N.: ERA5 hourly data on single levels from 1979 to present. Copernicus Climate Change Service (C3S) Climate Data Store (CDS). doi: 10.24381/cds.adbb2d47, 2018.

l.447 correct *dominatedby*
done

l.739 and the following. Please rephrase, since you do not *deliver* neither *accumulate* concentrations, as it is more properly written on l.744.
We substituted 'accumulation' by 'variation' and 'delivered' by 'released'. Also the sentences were put in a more logical order as follows:
*'At the long duration FAST station, the maximum net variations of NO3 and DIP concentrations within the ML before/after the rainy period reached i) 1520-665 = +855 µmol N m$^{-2}$ for NO3 (data from FAST+0.24 relative to FAST-0.25 and ii) 283-137 = +144 µmol P m$^{-2}$ for DIP (data from FAST+1.05 relative to FAST-0.25). In other terms, based on a mean MLD of 16 m, the net observed increase in the ML are + 9 nM DIP and + 54 nM NO3. As the rain event in the area would have increase by 0.07 nM DIP and 21 nM NO3 concentrations over the whole mixed layer, the net variations observed in the ML are thus higher than the calculated variation in stocks deduced from the N and P concentrations of this rain event (Table 3), in particular for DIP.'*

References
Pulido-Villena et al (2019) is cited on l.243 but is not included in the refs.
Yes this is an error the correct reference is Pulido-Villena et al 2010 which has been added in the ref list.

The following papers are in the References but are never cited in the text

Holmes, R. M., Aminot, A., Kérouel, R., Hooker, B. A,.and Peterson, B. J.: A simple and precise method for measuring ammonium in marine and freshwater ecosystems, Can. J. Fish. Aqua. Sci., 56, 1801-1808, 1999.
Ref removed

Karl, D. M.: Microbially mediated transformations of phosphorus in the sea: New views of an old cycle. Ann. Rev. Mar. Sci. 6: 279–337, doi: 10.1146/annurev-marine-010213-135046, 2014.
Ref removed

Krom, M. D., Herut, B., and Mantoura, R. F. C.: Nutrient budget for the eastern Mediterranean: Implication for phosphorus limitation, Limnol. Oceanogr, 49, 1582-1592, doi: 10.4319/lo.2004.49.5.1582, 2004.

This reference is now cited in the text in substitution to Krom et al., 2010

Krom, M. D., Emeis, K.-C., and Van Capellen, P.: Why is the eastern Mediterranean phosphorus limited? Progress In Oceanography, 85, 236-244, 2010.

Ref removed

Lindroth, P., and Mopper, K.: High performance liquid chromatographic determination of sub picomole amounts of amino acids by precolumn fluorescence derivatization with ophthaldialdehyde, Anal. Chem. 51, 1667–1674, 1979.

Ref removed

Thingstad, T. F., and Rassoulzadegan, F.: Nutrient limitations, microbial food webs, and 'biological C-pumps': suggested interactions in a P-limited Mediterranean, Mar. Ecol. Prog. Ser., 117, 299-306, 1995.

Ref removed

Zhang, J.-Z., and Chi, J.: Automated analysis of nano-molar concentrations of phosphate in natural waters with liquid waveguide, Environ. Sci. Technol., 36, 1048-1053, doi : 10.1021/es011094v, 2002.

The reference is now added in the text in the introduction section when referring to the LWCC technique

---

## Author Comment (AC2) · 9 Apr 2021

**Comments by referee 2 and our responses**

Comments on the manuscript bg-2020-411: Influence of atmospheric deposition on biogeochemical cycles in an oligotrophic ocean system

This manuscript by Van Wambeke et al. aims at investigating the impact of atmospheric depositions on the biogeochemical properties and processes in the Mediterranean Sea surface mixed layer. For this purpose, the authors present a large amount of data, collected both in the water column and in the atmospheric depositions, along a transect covering both the Western and Eastern basins of the Mediterranean Sea, during the PEACETIME cruise. The authors also presented the results of an enrichment experiment.

The strength of this paper, as the authors stated in their conclusions, is that it provides simultaneous sampling on both atmospheric depositions and the surface ocean on a large portion of the Mediterranean Sea. Moreover, as the authors highlighted in their introduction, the "in situ" study of atmospheric depositions is very rare due to obvious limitations. For these reasons, I think these data should be published.

However, the manuscript still needs a bit of refining. I hope that the following comments will be helpful to the authors.

We acknowledge the general comments of the referee and will respond to his/her more specific comments below. Citations of some parts of the revised version are in italics.

Main considerations:

1) I understand the difficulty of writing a paper with so many results. However, an effort need to be made to shorten the whole manuscript. In the current form I find it quite hard to read, it is too long, one can get lost while reading it. I suggest the authors try to smooth it and make it shorter. There are long sentences that can be shortened, or written with fewer words.

We agree, this was also recommended by the other referee. We considerably reduced the number of time we report values and ranges in the text, when those were available in a table. A new Table (S2) was added in the supplementary section to present N and C fluxes and stocks at the long duration stations ION and FAST.
The old paragraphs 2.3 in M&M and 3.5 in results section as well as old Figure 6 dealing on the enrichment experiments have been moved to the supplementary information (see below).
Section 3.3 has been reorganized in a more logical order; section 3.4 has been reduced, as well as the discussion. All in all, the ms is now reduced by a hundred lines (~2 pages).

2) I think that the setup of the enrichment experiment is somehow in contrast with the goal of this paper, which is studying the "in situ" effect of atmospheric deposition. As the authors highlighted in the introduction, these experiments are simplifying the natural system. How can we relate the results of an experiment carried out into a 60 ml bottle with what happens in the natural

environment? The authors themselves conclude that the results of the experiment cannot be compared with the "in situ" observations. I suggest removing this part.

We agree. The material and method, results and Figure describing the enrichment experiment has been moved to the supplementary section. We decided to keep it this way because the results of the enrichment experiments are cited in other ms of the special issue (Gazeau et al., 2021, Marañón et al., 2021). This enrichments experiment is now cited in the text only in the discussion phase (section 4.1) as follows:

*'The dependence of hprok on nutrients rather than on labile organic carbon during stratification conditions is not uncommon in the MS (Van Wambeke et al., 2002, Céa et al, 2017; Sala et al., 2002) and has been also shown during peacetime cruise (P, or N,P colimitation, Fig. S4).'*

3) It is clear that a lot of the results obtained from this cruise/project are presented in other papers that are currently under review in this issue or are being prepared.

There is a bit of confusion about some data, reported as results in this study, but at the same time citing other papers (under review or in preparation). In particular, I refer to the following:

Lines 418-421, DIP results in Pulido-Villena et al.

This manuscript is submitted this week. I can provide the ms draft on your request.

Lines 422, PP and BP in Maranon et al., 2020

This ms is now published and with free access:
https://bg.copernicus.org/articles/18/1749/2021/bg-18-1749-2021.pdf

Lines 432-435, LAP results in Van Wambeke et al., 2020

This ms is now published and with free access:
https://bg.copernicus.org/articles/18/2301/2021/bg-18-2301-2021.pdf

Lines 515-517, Citing the results from Fu et al.

The authors should clarify, if the results presented are already been present in the cited papers, they should be considered in the discussion section and not presented as results (and therefore also removed from the methods section). If this is the case, it would also help to make the whole manuscript shorter and more readable.

This part of the ms has been modified. As the two papers dealing with atmospheric deposition will be submitted likely in few weeks, we decided to provide in this revised version, all the information that are needed to understand atmospheric deposition estimation, i.e. details on the flux calculations from precipitation and nutrients analyses in the soluble fractions from aerosols samples in the M&M section, the estimations of N and P fluxes in rain samples. We also modified Table 3 accordingly.

Note that atmospheric wet deposition fluxes is not anymore only calculated from the time of collection and the diameter of rain funnel sampling rain on board, but rather from analysis of the soluble fraction of nutrients in the rain and precipitation estimates (in mm) from the radar data, and we refer to a published article for the methodology used (Hersbach et al., 2018). The M&M section was modified as: *'In the rains collected onboard the ship, NO3, NH4 and dissolved inorganic phosphorus (DIP) were also determined by ion chromatography after recovery of the dissolved fraction in the samples. Then, the wet deposition fluxes of these nutrients were estimated from the measured concentrations in the dissolved fractions of rains, multiplied by the*

*total precipitation. This total precipitation was issued from the hourly total precipitation accumulated during the rain event over the region from ERA5 hourly data reanalysis (Hersbach et al., 2018). Total precipitation was obtained by adding hourly rainfall on the grid-points (0.25° x 0.25°) spanning the ship location, more or less 1° around this central grid-point for integrating the regional variability (Table 3).'*

Hersbach, H., Bell, B., Berrisford, P., Biavati, G., Horányi, A., Muñoz Sabater, J., Nicolas, J., Peubey, C., Radu, R., Rozum, I., Schepers, D., Simmons, A., Soci, C., Dee, D., Thépaut, J-N.: ERA5 hourly data on single levels from 1979 to present. Copernicus Climate Change Service (C3S) Climate Data Store (CDS). doi: 10.24381/cds.adbb2d47, 2018.

4) There are way too many citations of articles that are "in preparation", I have counted at least 17. Citing a paper that is in preparation is usually not recommended (sometimes even not allowed), the data are not available and there is no guarantee that they will be. These citations need to be strongly reduced.

Yes, some of them were repeatedly cited in the first version of the ms and we should have paid more attention to that. We cited in the first version of the ms 6 papers in preparation, all intended to be submitted the special issue: Fu et al. b 5 times, Fu et al. a 1 time, Desboeufs et al. 4 times, Bressac et al. 3 times, Pulido-Villena et al. 6 times, Ridame et al. 1 time.

Since that time, Fu et al b and Desboeufs et al are merged in only one ms (Desbeoufs et al.), likely to be submitted before the end of April. This article as well as Fu et al. a are cited 'in prep for the special issue' but as previously said, the main information needed from these 2 papers are now reported in our revised version.

Bressac et al. in prep is not cited anymore

Pulido-Villena et al. has been submitted on the 09/04/2021:
Pulido-Villena, E., Desboeufs, K., Djaoudi, K., Van Wambeke, F., Barrillon, S., Doglioli, A., Petrenko, A., Taillandier, V., Fu, F., Gaillard, T., Guasco, S., Nunige, S., Triquet, S., and Guieu, C.: Phosphorus cycling in the upper waters of the Mediterranean Sea (Peacetime cruise): relative contribution of external and internal sources, Bigeosciences Discuss., this special issue, bg-2021-94, submitted

We don't cite anymore Ridame et al which was cited only once in the M&M section. We substituted this reference to Ridame et al. (2011), a reference which is already cited in the ms, as in this article the methodology is also described as well.

We can provide ms drafts to the referee upon request if necessary.

Minor comments:
Lines 169-172: This information is contained in table 1, they can be removed from the text and cite the table. Please also check that for stations ION and TYR the dates in the methods and table 1 do not correspond

Yes it is because on Table 1 we restricted time period to the CTDs casts sampling period, whereas in the text it was the whole ship time occupation of the sites. Anyway as suggested, we removed this paragraph in the text.

Lines 196 – 198: Define high and low frequency
At ION, it was 24 h, at FAST, the lower time lag between 2 CTDs was 7 h (at least those where both nutrients and BP were measured simultaneously). However, this part of the text was removed to reduce the ms.

Lines 405-416: This division in groups could be summarized in a table, to be more clear to the reader
The table already exists, it is Table S1. The paragraph was reduced.

Lines 425 – 427: Integrated PP and BP, how were they calculated? This information is missing in the methods
It was not written in this part but on line 379 of the previous version (the start of section 2.4) as follows:
'Trapezoidal integration was used to integrate BP, PP and N2fix within the ML.'
and later on in this section:
'Daily BP, AP and LAP integrated activities were calculated from hourly rates x 24'.

Paragraph 3.4 is a mixture of results and discussion and extra information that go beyond the results
We agree, this section was reduced to focus only on results.

---

## Author Response (AR1)

**Co-Editor-in-Chief Decision: Publish subject to minor revisions (review by editor)** (27 May 2021) by Christine Klaas

Comments to the Author:

Dear authors,

the associated editor recommended publication with minor revisions based on your answers to the reviewers. I am happy to accept this decision provided the manuscript has been revised accordingly including recommendations to improve the English.

Sincerely,

Christine Klaas

**Associate Editor Decision: Publish subject to minor revisions (review by editor)** (12 May 2021) by Chiara Santinelli

Comments to the Author (pdf): bg-2020-411-comments-to-author.pdf

Comments to the Author:

The authors carefully addressed most of the issues raised by the 2 referees. They modified the paper accordingly to the referee comments and suggestions. I think that the paper is improved a lot and it is almost ready for publication.

I added edits and comments in the attached file (please check they did not change the sense of the sentences); however, I recommend a further check of the English before final publication.

**Responses to Kristine Klass and Chiara Santinelli**

We thanks the Co editor in chief and the associate editor for their comments. The suggestions for correction of edits and comments have been taken into account. For this we used the ms corrected according our first responses to both referees.

Thus some comments made by the associate editor were sometimes already done, as this revised version was checked by an english native speaker.

To summarize, the final version of the ms in track change mode (including also the supplement section in track change mode) shows how the ms was reorganized to respond to both the referees as well as the associate editor.

---

## Editor Decision (ED1)

[revised manuscript text omitted]

Fig. 1

[Figure]

[Figure]

[Figure]

Fig.2

[Figure]

[Figure]

Fig. 3

[Figure]

[Figure]

[Figure]

Fig. 4

[Figure]

[Figure]

Fig. 5

[Figure]

[Figure]

Fig. 6

[Figure]

[Figure]

Fig. 7

---

## Author Response (AR2)

**Co-Editor-in-Chief Decision:**

Christine Klass, 1st July 2021

Unfortunately the language (English) of the manuscript does not have the acceptable standard for publication. In the time available before submitting my decision I revised the manuscript and made suggestions for improvements up to p. 10 (see annotated manuscript). I urge you to thoroughly check the remaining text before resubmission.

Sincerely,

Christine Klaas

**Responses** We thank Christine Klass for her corrections. The suggestions for correction of edits and comments have been taken into account.

To be clearer for the reader, the limit of quantification has been defined in the text (calculated as ten times the standard deviation of ten measurements of the blank), this allows to differentiate from limit of detection (3 times the sd of 10 measurements of the blank).

Since both quantification and quantitation have the same meaning, we prefer to keep the term quantification.

---

## Editor Decision (ED2)

[revised manuscript text omitted]

NO3 Aerosol [μg m⁻³]

Fig. 1

[Figure]

Fig.2

[Figure]

Fig. 3

[Figure]

Fig. 4

[Figure]

Fig. 5

[Figure]

Fig. 6

---

## Editor Decision (ED3)

[revised manuscript text omitted]

Fig. 1

[Figure]

Fig.2

[Figure]

Fig. 3

[Figure]

Fig. 4

[Figure]

Fig. 5

[Figure]

Fig. 6

---

## Author Response (AR4)

**Co-Editor-in-Chief Decision:**
Christine Klass, 29 Jul 2021

**Responses in blue**

Dear authors,
The text of the manuscript has been greatly improved, however, in particular in the results section, the text warrants further polishing (comments are given in the annotated manuscript).

We warmly thank you for these comments and we have accounted for all your corrections except one:

Lines 405-406 of the pdf: Substitution of 'contributed' by 'represented' . We prefer to keep 'represented'. The term contribution would imply a direct utilization of organisms to satisfy their N or P requirements, which is not sure. The flux could be direct or indirect after circulation through the microbial food web. Also, ectoenzymatic hydrolysis and N2 fixation can fuel different categories of microorganisms (phytoplankton and heterotrophic prokaryotes).

Response to your comment line 335 of the pdf:
"At the short stations and sites, the term $NO3_{ML} - NO3_{NDLb}$ can be inferred by the difference between mean nanomolar (LWCC) concentrations within the NDLb and the ML, as advective fluxes could not be characterized "
Why not please explain?

The sentence was modified to be more explicit as :
*'At short stations, only single casts were carried out, preventing any estimation of temporal variations of the MLD (dMLD/dt) required for the calculation of vertical advective fluxes. Thus only a qualitative assessment of nutrient fluxes across ML is given'.*

Response to your comment line 469 of the pdf:
"This does not correspond to the data on NO3 fluxes in Table S2. Can you explai  the difference?"
Yes we apologize for this error. Table S2 presents the DIN (NH4+NO3) and in the text we described NO3 only. We modified table S2 and text to talk only about DIN.

I also noticed one issue that should be clarified:

- Lines 445-448 of the pdf: I fail to understand the argument here. In the text before increases in windspeed associated with rain events are presented as driving MLD and NO3 increases between the 25th and 27th of May; while here the authors shift to invoque dry deposition without presenting any quantitative basis. By looking at Fig. 4 average ML NO3 concentrations increase from ~20nM to ~125nM. This implies that N stocks in the upper 20m of the water column increase from 400 µMol/m2 to 2500 µMol/m2 necessitating an N input of 2100 µMol/m2 (2.1 mmol/m2).

We agree with your comments, the exact values are noted Table S1
On 25 May , MLD 14 m, mean NO3 in ML 14 nM, in NDLb 14 nM
On 27 May  MLD 18 m, man NO3 in ML 127 nM, in NDLb 103 nM
On 29 May MLD 16 m, mean NO3 in ML 92 nM, in NDLb 67 nM

We also calculated NO3 integrated stocks in the ML (cited on Table S2) : On 25 May: 195 µmole/m2, on 27 may: 2113 µmole/m2, i.e., a net difference of + 1918 µmole/m2 in  2.12 days between the 2 casts, confirming your estimates.

Based on the deposition values given in table 3 and table S2, it seems clear that neither of these (wet or dry) terms can explain these changes in such a short period of time.

We agree, dry deposition: at best 32.6 µmole/m2/d (on 27 May ) x 2.12 = 69 µmole N/m2 in 2.12 days, N2 fixation at best 6.1 µmole/m2/d x 2.012 = 12.9 µmole/m2 so we agree that the sum of these 2 fluxes are not sufficient to justify such increase in the ML. The wet deposition flux cited Table 3 (67 µmole NO3 /m2) corresponds to the rain front of the 29 May and should not be considered here).

This is compounded by the fact that these changes actually occur throughout the water column including below the NDLb. Do you have an explanation? Could it be that different water masses were sampled at ION?

Yes, it is clear that both ML and NDLb are enriched in NO3. The main problem is that the CTD casts are not sufficiently close to allow us to calculate subtle variations in dMLD/dt together with NO3 vertical gradients. As we discuss later for FAST, these can be temporary very important to justify intermittently large source of exchange of NO3 between ML and NDLb. Furthermore, at ION we have only 3 nitrate profiles, and the second one (on 27 May) is not sufficiently deep (it stops at 45m) to see any change in the form of the deep nitracline below NDLb.
The changes in water mass properties remain small below the MLD of all the casts done between 25 and 27 May (see the figure below). Thus we could not conclude on a significant vertical mixing nor to a water mass change.

[Figure]

The thermosalinograph allowed us to get continuous information of the surface water together with the wind. The wind is plotted on Figure 3 of the ms and we present below a detailed view of the wind, and surface salinity between the 25 and the 27 May , with the 2 casts sampled for nutrients indicated in red:

[Figure]

On this graph, the wind stress of the 27 May is concomitant to a lowering of 0.1 unit of the surface salinity, which is also visible on the 27 May cast down to about 10 m depth. Thus we could not exclude the possibility of an intrusion of a low-salinity surface water lens. This salinity decrease corresponds approximatively to a rainfall of 20 liters over a layer of 10 m, corresponding to a precipitation of 20 mm i.e. much higher than the rain event of the 29 May cited Table 3.

A possible scenario could be the following: a low-salinity lens could have its origin from the rain events observed in the ship's vicinity on the 26 May. This lens could be advected via Ekman transport associated to the wind stress to progressively influence the station location.

The sentence in the ms was modified as (lines 620-624 of the revised word doc in 3.3) :
*Due to the lack of high frequency sampling, it was not possible to quantitatively assess the effects of dry /wet atmospheric deposition nor the one of nitrate injection from below the NDLb by vertical advection at ION. The intrusion of a low-salinity lens was clearly visible on the thermosalinograph record and on the 27 May CTD cast, extending down to 10 m depth (data not shown). This low-salinity lens could be formed by the rain event noted on 26 May in the vicinity of the station. It was clear that ION on days 27 and 29 was characteristic of group 4 (i.e. higher NO3 concentrations in the ML than in the NDLb), presumably related to NO3 rainfall inputs.*

Also, in lines 731-732 of the pdf you mention "considering both the local rain fluxes and the horizontal oceanic mixing of water masses affected by the rain front". However, horizontal mixing was not considered here. Can you clarify what is meant by this?

You are right, the concept of "horizontal mixing" in this sentence in conclusion is confusing. We

meant that for either dry deposition directly measured or wet deposition indirectly inferred during long stations, their signature was detected in sea water nutrient concentrations. As you can see Table 3, the nutrients in rain were analyzed in the rain sampled on board, whereas the whole rain flux was derived from mean precipitation estimated from radar data in the vicinity of the ship's position.

We removed this sentence.

---

## Editor Decision (ED4)

[revised manuscript text omitted]

Fig. 1

[Figure]

Fig.2

[Figure]

Fig. 3

[Figure]

Fig. 4

[Figure]

Fig. 5

[Figure]

Fig. 6